# Attention Normalization Impacts Cardinality Generalization in Slot Attention

**Markus Krimmel**                                                    *markus.krimmel@tuebingen.mpg.de*
*Embodied Vision Group, Max Planck Institute for Intelligent Systems*

**Jan Achterhold**                                                      *jan.achterhold@tuebingen.mpg.de*
*Embodied Vision Group, Max Planck Institute for Intelligent Systems\**

**Joerg Stueckler**                                                        *joerg.stueckler@uni-a.de*
*Embodied Vision Group, Max Planck Institute for Intelligent Systems*
*Intelligent Perception in Technical Systems Group, University of Augsburg*

**Reviewed on OpenReview:** *https://openreview.net/forum?id=llQXLfbGOq*

## Abstract

Object-centric scene decompositions are important representations for downstream tasks in fields such as computer vision and robotics. The recently proposed Slot Attention module, already leveraged by several derivative works for image segmentation and object tracking in videos, is a deep learning component which performs unsupervised object-centric scene decomposition on input images. It is based on an attention architecture, in which latent slot vectors, which hold compressed information on objects, attend to localized perceptual features from the input image. In this paper, we demonstrate that design decisions on normalizing the aggregated values in the attention architecture have considerable impact on the capabilities of Slot Attention to generalize to a higher number of slots and objects as seen during training. We propose and investigate alternatives to the original normalization scheme which increase the generalization capabilities of Slot Attention to varying slot and object counts, resulting in performance gains on the task of unsupervised image segmentation. The newly proposed normalizations represent minimal and easy to implement modifications of the usual Slot Attention module, changing the value aggregation mechanism from a weighted mean operation to a scaled weighted sum operation.

## 1 Introduction

Object-wise scene decompositions are ubiquitous in computer vision, robotics, and related disciplines such as reinforcement learning, since the state and actions in the environment are naturally represented in relation to objects. Over recent years, unsupervised learning of object-centric representations from unlabelled images and video has attracted significant interest in the machine learning community (Greff et al., 2017; Engelcke et al., 2020; Locatello et al., 2020). The Slot Attention architecture (Locatello et al., 2020) decomposes a two-dimensional RGB input image object-wise using an attention mechanism which updates slots, holding information about objects, in a recurrent manner. In (Locatello et al., 2020), it was used for set prediction and image segmentation tasks on relatively simple renderings of 2D or 3D scenes, such as CLEVR (Johnson et al., 2017). Later in (Seitzer et al., 2023), Slot Attention has also been successfully applied to the task of unsupervised segmentation of more realistic images (MOVi, (Greff et al., 2022)). Detecting and tracking objects in videos using Slot Attention is described in (Kipf et al., 2022; Elsayed et al., 2022). Despite its empirical success, a theoretical explanation of its inner workings and the inductive biases which lead to

---

*\*Work done during PhD studies at MPI-IS. Jan Achterhold is now with Robert Bosch GmbH - Corporate Research, and the Bosch Center for Artificial Intelligence (BCAI), Renningen, Germany.*

the emergence of object-wise decompositions in Slot Attention is still under active research (Chang et al., 2022b;a).

In this paper we investigate design choices on the *normalization of aggregated attention values* in Slot Attention. We find that the normalization proposed in (Locatello et al., 2020) leads to suboptimal foreground segmentation performance during inference with higher number of objects or slots than used for training the model. We also investigate two alternative normalization approaches, give theoretical insights on their behavior, and assess their performance in relation to the original Slot Attention baseline. We demonstrate that these different approaches for normalizing the aggregated values can have a significant impact on the generalization of Slot Attention to a varying number of slots and objects during inference.

## 2 Background

### 2.1 Slot Attention

The Slot Attention module (Locatello et al., 2020) is given a set of $N$ input tokens $\tilde{\boldsymbol{x}}_n \in \mathbb{R}^{D_{\text{input}}}, n \in \{1, ..., N\}$ and iteratively refines a set of $K$ slots $\tilde{\boldsymbol{\theta}}_k \in \mathbb{R}^{D_{\text{slot}}}, k \in \{1, ..., K\}$. In an object-wise scene decomposition scenario, slots correspond to latent variables holding information on objects, while the input tokens are localized image features, e.g., computed by a convolutional neural network. Slots bind to input tokens via a dot-product attention mechanism (Luong et al., 2015). Learned linear maps $k$ and $q$ extract $D$-dimensional keys and queries from the layer-normalized (Ba et al., 2016) input tokens $\boldsymbol{x}_n := \text{LayerNorm}(\tilde{\boldsymbol{x}}_n)$ and layer-normalized slots $\boldsymbol{\theta}_k := \text{LayerNorm}(\tilde{\boldsymbol{\theta}}_k)$, respectively. In our case, we always have $D_{\text{input}} = D$ and the layer normalization modules that produce $\boldsymbol{x}_n$ and $\boldsymbol{\theta}_k$ do not share parameters.

For the attention mechanism, an unnormalized $N \times K$ matrix $\boldsymbol{M}$ of dot products is formed from the keys $\boldsymbol{k}_n := k(\boldsymbol{x}_n)$ and queries $\boldsymbol{q}_k := q(\boldsymbol{\theta}_k)$. On each row of $\boldsymbol{M}$, a Softmax operator is then applied, yielding $\boldsymbol{\Gamma} = (\gamma_{n,k}) \in [0,1]^{N \times K}$:

$$M_{n,k} := \frac{1}{\tau} k(\boldsymbol{x}_n)^\top q(\boldsymbol{\theta}_k) = \frac{1}{\tau} \boldsymbol{k}_n^\top \boldsymbol{q}_k \qquad (1) \qquad\qquad \gamma_{n,k} := \frac{\exp M_{n,k}}{\sum_{k'=1}^{K} \exp M_{n,k'}}. \qquad (2)$$

With this, each row $\boldsymbol{\gamma}_{n,:}$ may be interpreted as the probability of an input token $n$ to be assigned to a particular slot $k$. The constant $\tau$ corresponds to a temperature parameter which is chosen to be $\sqrt{D}$.

A linear map $v : \mathbb{R}^{D_{\text{input}}} \to \mathbb{R}^D$ extracts values from the input tokens and the matrix $\boldsymbol{\Gamma}$ is used to accumulate values into unnormalized slot-wise update codes:

$$\tilde{\boldsymbol{u}}_k := \sum_{n=1}^{N} \gamma_{n,k} v(\boldsymbol{x}_n) \qquad (3)$$

With the motivation to improve the stability of the attention mechanism, Slot Attention performs a normalization on the update codes. Namely, the sum in (3) is scaled in such a way that it becomes a weighted mean of the values $v(\boldsymbol{x}_n)$, i.e.:

$$\boldsymbol{u}_k := \frac{\tilde{\boldsymbol{u}}_k}{\sum_{n=1}^{N} \gamma_{n,k}} \qquad (4)$$

This normalization scheme is termed *weighted mean*. Locatello et al. (2020) discuss two ablations of this normalization. The *weighted sum* scheme normalizes the update code by multiplication with a constant, i.e. $\boldsymbol{u}_k := \frac{1}{C} \tilde{\boldsymbol{u}}_k$. In the ablation study in (Locatello et al., 2020), the value chosen for $C$ is not discussed, and it must be assumed that $C = 1$ was chosen. The second ablation of (Locatello et al., 2020) is termed *layer normalization* and uses a layer normalization module that is shared across slots for normalization. Concretely, the normalized update code is computed as $\boldsymbol{u}_k := \text{LayerNorm}(\tilde{\boldsymbol{u}}_k)$. We refer to Appendix B for an exact definition of layer normalization.

For each slot $k$, the aggregated value $\boldsymbol{u}_k$ is used to update the latent representation $\tilde{\boldsymbol{\theta}}_k$ via a gated recurrent unit (GRU) (Cho et al., 2014) and a residual multilayer perceptron with $\tilde{\boldsymbol{\theta}}_k^{\text{new}} := \texttt{update}(\tilde{\boldsymbol{\theta}}_k, \boldsymbol{u}_k)$.

## 2.2 Von Mises-Fisher Distributions

Von Mises-Fisher (vMF) distributions (Fisher, 1953) are probability distributions on the unit $(d-1)$-sphere in $\mathbb{R}^d$. Typically, they are parametrized by a mean direction $\boldsymbol{\theta} \in \mathbb{R}^d$ with $\|\boldsymbol{\theta}\|_2 = 1$ and a concentration parameter $\tau > 0$. They are defined by the following density w.r.t. the usual surface measure on the $(d-1)$-sphere $f(\boldsymbol{x} \mid \boldsymbol{\theta}, \tau) = \frac{1}{Z(d,\tau)} \exp\left(\frac{\boldsymbol{\theta}^\top \boldsymbol{x}}{\tau}\right)$, where $Z(d,\tau)$ is a normalization constant that is independent of $\boldsymbol{\theta}$. If $(\boldsymbol{\theta}_1, ..., \boldsymbol{\theta}_K)$ and $(\tau_1, ..., \tau_K)$ are parameters of vMF distributions and $(\pi_1, ..., \pi_K)$ is contained in the probability simplex, a vMF mixture model can be defined as usual via the density $g(\boldsymbol{x}) := \sum_{k=1}^{K} \pi_k f(\boldsymbol{x} \mid \boldsymbol{\theta}_k, \tau_k)$.

## 3 Slot Attention and von Mises-Fisher Mixture Model Parameter Estimation

Many works (Locatello et al., 2020; Chang et al., 2022b;a; Kirilenko et al., 2023) compare Slot Attention to expectation maximization (Dempster et al., 1977; Bishop, 2006) (EM) in Gaussian mixture models, i.e. to soft k-means clustering. We, however, connect it with expectation maximization in a mixture model of von Mises-Fisher (vMF) distributions (Banerjee et al., 2003), since Slot Attention uses a bilinear form on slots and inputs as a scoring function instead of the negative Euclidean distance. In this section, we make the parallel between Slot Attention and EM explicit by performing EM parameter estimation in a vMF mixture model and relating each step to the corresponding step in Slot Attention. We will then view the weighted mean, layer norm, and weighted sum normalization variants in the context of this analogy and compare them.

### 3.1 Relating Slot Attention to EM

We consider a case in which $N$ points $\boldsymbol{x}_n$ are given on the unit $(d-1)$-sphere in $\mathbb{R}^d$. We estimate the mean directions $\boldsymbol{\theta}_1, ..., \boldsymbol{\theta}_K$ of $K$ vMF components, along with the mixture coefficients $\pi_1, ..., \pi_K$. We assume that the vMF distributions have fixed concentration, i.e., $\tau = 1$. We interpret the parameters $\boldsymbol{\theta}_k$ to relate to slots in Slot Attention and the points $\boldsymbol{x}_n$ to relate to the perceptual input features of the module. The concentration parameter $\tau$ can be understood as an analogue to the temperature $\sqrt{D}$ in Slot Attention.

**E-Step** In the expectation step, soft assignments of datapoints to clusters (slots) are computed via the likelihood functions of the vMF components:

$$\gamma_{n,k} := \frac{\pi_k \exp(\boldsymbol{x}_n^\top \boldsymbol{\theta}_k)}{\sum_{k'=1}^{K} \pi_{k'} \exp(\boldsymbol{x}_n^\top \boldsymbol{\theta}_{k'})} \tag{5}$$

The resulting matrix $\boldsymbol{\Gamma} \in [0,1]^{N \times K}$ corresponds to the attention matrix in Slot Attention. Equation (5) closely resembles the computation of the attention matrix in Slot Attention with some differences: While the inputs $\boldsymbol{x}_n$ in Slot Attention do not necessarily lie on the unit sphere, we do remind the reader that they are layer-normalized and therefore lie on ellipsoids. Similarly, the slots are layer-normalized before the attention step. In contrast to equation 5, Slot Attention uses key and query maps instead of directly forming a dot product between $\boldsymbol{x}_n$ and $\boldsymbol{\theta}_k$. I.e., the dot products are formed between $\boldsymbol{k}_n$ and $\boldsymbol{q}_k$.

While the Slot Attention architecture does not explicitly model the mixture parameters $\pi_k$, it may encode some weighting in the layer-normalized slots. Indeed, we show in Appendix B that the keys $\boldsymbol{k}_n$ are contained in some $(D-1)$-dimensional affine subspace $A \subsetneq \mathbb{R}^D$ which may be written uniquely as $A = \boldsymbol{a} + V$ where $V$ is a $(D-1)$-dimensional linear space and $\boldsymbol{a} \in V^\perp$ is perpendicular to $V$. If $p_V : \mathbb{R}^D \to V$ is the orthogonal projection onto $V$ and $p_a : \mathbb{R}^D \to \langle \boldsymbol{a} \rangle$ is the orthogonal projection onto the span of $\boldsymbol{a}$, we may decompose any $\boldsymbol{x} \in \mathbb{R}^D$ orthogonally as $\boldsymbol{x} = p_V(\boldsymbol{x}) + p_a(\boldsymbol{x})$. For any key vector $\boldsymbol{k}_n = k(\boldsymbol{x}_n) \in A$ we therefore have $\boldsymbol{k}_n = \boldsymbol{a} + p_V(\boldsymbol{k}_n)$. The attention value $\gamma_{n,k}$ in Slot Attention may now be written as:

$$\gamma_{n,k} = \frac{\exp(\boldsymbol{k}_n^\top \boldsymbol{q}_k)}{\sum_{k'} \exp(\boldsymbol{k}_n^\top \boldsymbol{q}_{k'})} = \frac{\exp(\boldsymbol{a}^\top p_a(\boldsymbol{q}_k)) \exp(p_V(\boldsymbol{k}_n)^\top p_V(\boldsymbol{q}_k))}{\sum_{k'} \exp(\boldsymbol{a}^\top p_a(\boldsymbol{q}_{k'})) \exp(p_V(\boldsymbol{k}_n)^\top p_V(\boldsymbol{q}_{k'}))} \tag{6}$$

Hence, the term $\exp(\boldsymbol{a}^\top p_a(\boldsymbol{q}_k))$ may be interpreted as an analogue of $\pi_k$, which assigns a weight to the $k^{\text{th}}$ slot but is independent of the input at index $n$.

**M-Step** In the maximization step, cluster (slot) parameters are updated using the soft assignments $\mathbf{\Gamma}$. In EM, the new mean directions and mixing coefficients are computed via:

$$\boldsymbol{\theta}_k^{\text{new}} := \frac{\sum_{n=1}^N \gamma_{n,k} \boldsymbol{x}_n}{\left\| \sum_{n=1}^N \gamma_{n,k} \boldsymbol{x}_n \right\|_2} \qquad (7) \qquad\qquad \pi_k^{\text{new}} := \frac{\sum_{n=1}^N \gamma_{n,k}}{N} \qquad (8)$$

In our analogy to Slot Attention, this M-step would relate to the slot-update involving the aggregated values $\boldsymbol{u}_k$. Hence, it may be of interest in this comparison to investigate whether the values $\boldsymbol{u}_k$ hold information about the right-hand sides of equations (7) and (8).

### 3.2 Comparing Normalizations in EM Analogy

In the following paragraphs, we discuss how the update normalizations discussed previously compare in the context of our EM analogy and, in particular, whether the normalized update codes $\boldsymbol{u}_k$ hold sufficient information to recover the quantities from equations (7) and (8).

**Weighted Mean** In the weighted mean case, the aggregated values $\boldsymbol{u}_k$ can hold sufficient information to extract the quantities in (7) and (8) if $D = D_{\text{input}}$ holds. Assuming that the value map is the identity, the right-hand side of (7) may be computed as $\boldsymbol{u}_k / \|\boldsymbol{u}_k\|_2$. However, it is not clear how $\pi_k$ could be computed from $\boldsymbol{u}_k$. Indeed, we show in Proposition 1 that there can be no general formula as for the weighted sum case that generalizes without exception across slot-counts. We provide a proof in Appendix C by constructing some explicit slot settings which demonstrate that a hypothetical function $f$ can not map every update code to a corresponding unique scalar.

**Proposition 1.** *Consider Slot Attention with weighted mean normalization and any fixed model parameters and fixed input data $\tilde{\boldsymbol{x}}_1, ..., \tilde{\boldsymbol{x}}_N$ with $N \geq 1$. Then, there exists no function $f : \mathbb{R}^D \to \mathbb{R}$ such that it holds*

$$f(\boldsymbol{u}_k) = \frac{\sum_{n=1}^N \gamma_{n,k}}{N} \qquad \forall 1 \leq k \leq K \qquad (9)$$

*for arbitrary $K \geq 1$, arbitrary slots $\tilde{\boldsymbol{\theta}}_1, ..., \tilde{\boldsymbol{\theta}}_K$ and resulting normalized update codes $\boldsymbol{u}_1, ..., \boldsymbol{u}_k$.*

**Layer Normalization** While, at least in some cases, it is possible to recover the quantity in (7) from update codes in the layer-normalization variant, these update codes still do not contain sufficient information to infer the quantity in (8). Indeed, the reader may verify that the same argument we presented in the proof of Proposition 1 also holds for the layer norm variant.

**Weighted Sum** In the weighted sum case, we may, as for the weighted mean normalization, obtain the right-hand side of equation (7) via $\boldsymbol{u}_k / \|\boldsymbol{u}_k\|_2$ if the value map is the identity. In contrast to the previously discussed normalizations, we may also recover information on the column sums $\sum_{n=1}^N \gamma_{n,k}$, which appear in equation (8). We make this rigorous in Proposition 2 and provide a proof in Appendix D, where we exploit the fact that the values $\boldsymbol{v}_n$ lie in a lower-dimensional subspace.

**Proposition 2.** *Consider Slot Attention with weighted sum normalization and fixed model parameters. Let the number of input tokens $N$ be fixed. Assume that $D_{input} = D$ holds. For almost all (w.r.t. Lebesgue measure) parameters of the input's layernorm module and the value map $v$, there exists a map $f : \mathbb{R}^D \to \mathbb{R}$ (which may depend on these parameters) such that*

$$f(\boldsymbol{u}_k) = \frac{\sum_{n=1}^N \gamma_{n,k}}{N} \qquad \forall 1 \leq k \leq K \qquad (10)$$

*holds for any $K \geq 1$, any slots $\tilde{\boldsymbol{\theta}}_1, ..., \tilde{\boldsymbol{\theta}}_K$, any input data $\tilde{\boldsymbol{x}}_1, ..., \tilde{\boldsymbol{x}}_N$, and the resulting attention matrix $\mathbf{\Gamma}$ and update codes $\boldsymbol{u}_1, ..., \boldsymbol{u}_K$.*

Since the assumptions of Proposition 2 only exclude a parameter subset of zero volume, its conclusion likely holds in practice during training.

Hence, the weighted sum variant may also be seen as a generalization of the weighted mean normalization: the update codes from weighted sum normalization hold sufficient information such that weighted-mean-normalized update codes can be recovered from them (e.g. by the `update` network if it has sufficient capacity). The reverse is not true, as demonstrated in Proposition 1.

## 4  Methods of Normalization

### 4.1  Weighted Sum Normalization with Fixed Scaling

As detailed in the above discussion, in contrast to the weighted mean normalization, the weighted sum normalization may preserve information on the fraction $\pi_k$ of input tokens assigned to the slot in the slot update code $\boldsymbol{u}_k$. While (Locatello et al., 2020) report worse performance of the weighted sum normalization compared to the weighted mean normalization, the value chosen for $C$ is not further discussed. As detailed in our experiments, we observe that the weighted sum normalization can outperform the weighted mean normalization for $C = N$, where $N$ is the number of input tokens. For image inputs, the number of tokens is relatively large, e.g. $N = 128^2 = 16,384$ for feature maps of CLEVR renderings. We aim to avoid unreasonably large values in the update code $\boldsymbol{u}_k$, which may lead to numerical instabilities, such as vanishing gradients. With $C = N$, it holds that $\boldsymbol{u}_k$ is bounded with $|u_{k,d}| \leq \max_n |v_{n,d}| \ \forall d \in \{1, ..., D\}$, which is independent of $N$. Indeed, we have the following chain of inequalities:

$$|u_{k,d}| = \left| \frac{1}{N} \sum_{n=1}^{N} \gamma_{n,k} v_{n,d} \right| \leq \frac{1}{N} \sum_{n=1}^{N} \gamma_{n,k} |v_{n,d}| \leq \frac{1}{N} \sum_{n=1}^{N} \gamma_{n,k} \max_n |v_{n,d}| \leq \max_n |v_{n,d}| \tag{11}$$

Here, we first use the triangle inequality, followed by the crude estimate $|v_{n,d}| \leq \max_n |v_{n,d}|$ for all $n$. Finally, we used the fact that $\sum_{n=1}^{N} \gamma_{n,k} \leq N$ holds, since $\boldsymbol{\Gamma}$ is row-stochastic with $N$ rows. While this provides an argument for our choice of $C$ which is a suitable heuristics across tasks, we hypothesize that task-specific tuning of this hyperparameter may be beneficial.

### 4.2  Weighted Sum Normalization with Batch Scaling

Instead of heuristically choosing a scaling parameter $C$ in the weighted sum normalization as above, we also investigate an approach in which the scaling factor is learned via a form of batch normalization (Ioffe & Szegedy, 2015) during training. Concretely, we measure the magnitude of unnormalized update vectors in the *first Slot Attention iteration of each forward pass* by computing their batch statistics. These batch statistics are used during the subsequent iterations of the forward pass to scale the update codes. While other works using batch normalization in recurrent networks do not share statistics across time (Cooijmans et al., 2017; Laurent et al., 2016), we find that our approach greatly simplifies varying the number of iterations during inference. In contrast to typical implementations of batch normalization, we propose to reduce all axes during the computation of the statistics (i.e., the batch axis, the slot axis, and the layer axis). Reducing the slot axis is necessary to preserve slot-permutation equivariance, which is a desirable property in object-centric learning (Locatello et al., 2020). Reducing the layer axis leads to scalar batch statistics, yielding a method that more closely aligns with the normalization approaches we have discussed so far.

Assuming that the tensor $\tilde{\boldsymbol{U}}^{(0)} \in \mathbb{R}^{L \times K \times D}$ holds the unnormalized update codes of the first SA iteration computed for a mini-batch of size $L$, we define the batch statistics as:

$$m := \frac{1}{LKD} \sum_{l=1}^{L} \sum_{k=1}^{K} \sum_{i=1}^{D_{\text{slot}}} \tilde{U}_{l,k,i}^{(0)} \qquad v := \frac{1}{LKD-1} \sum_{l=1}^{L} \sum_{k=1}^{K} \sum_{i=1}^{D_{\text{slot}}} (\tilde{U}_{l,k,i}^{(0)} - m)^2 \tag{12}$$

Note that both statistics are scalar-valued. As proposed by Ioffe & Szegedy (2015), we also learn two parameters $\alpha, \beta \in \mathbb{R}$ and normalize the tensor of update codes in iteration $j$ via:

$$\boldsymbol{U}^{(j)} := \alpha \frac{\tilde{\boldsymbol{U}}^{(j)} - m}{\sqrt{v + \epsilon}} + \beta \tag{13}$$

where $\epsilon > 0$ is a small constant. We stress that the values $m$ and $v$ are computed for each mini-batch in the first Slot Attention iteration and are therefore independent of $j$. Moreover, gradients flow through $m$ and $v$. We cache an exponential moving average of the batch statistics during training and use it during inference. Hence, during inference, the normalization in equation (13) is simply an affine transformation with fixed weights, thereby closely resembling the weighted sum normalization presented in the previous subsection. While the weighted sum normalization is linear and not affine, we note that this distinction does not impact the capacity of the model. Indeed, the vectors $\boldsymbol{u}_k$ are also affinely transformed within the `update` network, therefore making any previous affine or linear transformation redundant from a capacity perspective. Notwithstanding, the normalizations presented here will significantly alter the trainig trajectory of the models. Batch normalization, in particular, has previously been shown to remedy the problem of saturating activations and vanishing gradients (Ioffe & Szegedy, 2015; Pascanu et al., 2013), which may arise from improper normalization (Glorot & Bengio, 2010). We provide pseudocode for the two proposed normalization variants in Appendix G.

## 5 Experiments

We investigate the proposed normalizations on unsupervised object discovery tasks. To this end, we train autoencoders on the CLEVR (Johnson et al., 2017) and MOVi-C (Greff et al., 2022) datasets, utilizing autoencoder architectures that have been described in (Locatello et al., 2020) and (Seitzer et al., 2023), respectively. Additional results for a property prediction task can be found in Appendix I. We provide visualizations of scene segmentations in Appendix H. We will give a brief overview on our experimental setup in the following and refer to the supplementary material for more details[1].

**Model Variants**   We refer to the standard normalization (weighted mean) as the *baseline* and to the LayerNorm-based ablation from (Locatello et al., 2020) as the *layer* normalization. We term the method detailed in Sec. 4.1 the *weighted sum* normalization, and the method from Sec. 4.2 the *batch* normalization. In some experiments, we will train models on filtered training sets (CLEVR6 and MOVi-C6) containing only a limited number of objects. For clarity, we annotate each model variant with a tuple $(O, K)$, where $O$ denotes the maximum number of objects seen in the training set and $K$ denotes the number of slot latents used during training.

**CLEVR Dataset**   We use an extended version of the CLEVR dataset that is provided in the Multi-object Datasets repository (Kabra et al., 2019). It consists of 100,000 2D renderings of 3D scenes depicting up to 10 objects whose shapes are geometric primitives. Each scene is annotated with a ground truth segmentation, which we use for evaluation. Following Locatello et al. (2020), we use 70,000 images for training and further adopt the approach of (Locatello et al., 2020; Greff et al., 2019; Burgess et al., 2019) by cropping the images to highlight objects in the center. In contrast to (Locatello et al., 2020), we also augment the data during training via random horizontal flips. As in (Locatello et al., 2020), we also consider a subset of the CLEVR dataset, only consisting of images containing at most 6 objects. We refer to this dataset as CLEVR6 and will denote the original dataset by CLEVR10.

**MOVi-C Dataset**   Compared to CLEVR, MOVi-C represents a significant step-up in perceptual complexity. It contains 10,986 video sequences, each consisting of 24 frames. We use 250 of these video sequences for validation and hold out 999 sequences for testing. Each clip shows 3 to 10 highly textured 3D-scanned objects from the Google Scanned Objects repository (Downs et al., 2022) flying into view and colliding. In our experiments, we only consider single RGB frames from the dataset and discard any temporal relation between them. Once again, we introduce a filtered dataset, which we denote by MOVi-C6 and which consists of frames of clips that contain at most 6 objects. For sake of clarity, we refer to the original dataset as MOVi-C10.

**MOVi-D Dataset**   The MOVi-D dataset consists of scenes that are visually similar to those from the MOVi-C dataset. However, the scenes contain up to 23 (10 to 20 static, 1 to 3 moving) objects, thereby presenting a greater challenge to object-centric method. Structurally, the dataset resembles MOVi-C, consisting of 11,000

---

[1]Code is available at `https://github.com/EmbodiedVision/slot_attention_normalization`.

video sequences, each made up of 24 frames. As before, we discard any temporal relationship between frames. In our experiments we will not train on MOVi-D, but instead investigate zero-shot transfer performance of models that were trained on MOVi-C.

**CNN Autoencoder** We adopt the convolutional neural network (CNN) based architecture proposed in (Locatello et al., 2020) to train object-centric autoencoders on the CLEVR dataset. A convolutional network transforms input images into feature maps, which are enriched by positional embeddings and spatially flattened. The resulting sets of tokens are processed by the Slot Attention module to obtain object-centric latent representations. A spatial broadcast decoder (Watters et al., 2019) decodes each slot latent separately into an image and an unnormalized alpha mask. The alpha masks are normalized across the slot axis via a softmax operation and subsequently used to linearly combine the reconstructed images, thereby producing a reconstruction of the input. As in (Locatello et al., 2020), we perform 3 Slot Attention iterations during training, and 5 iterations during evaluation. We further follow (Locatello et al., 2020) in that we obtain segmentations from trained autoencoders by assigning each pixel to the slot for which the corresponding entry in the alpha mask attains a maximum value.

**Dinosaur Autoencoder** To obtain object-centric behavior on the substantially more complex MOVi-C dataset, we adopt the approach of Dinosaur (Seitzer et al., 2023). Instead of directly operating on RGB frames, the autoencoders are trained on image features that are extracted via a pre-trained and fixed vision transformer (ViT) (Caron et al., 2021). In spirit, the autoencoder resembles the previously discussed architecture: A small encoder, in the form of a two-layer perceptron, processes the ViT features, which are then transferred into a latent representation by the Slot Attention module. Each latent is decoded individually into a reconstruction of the image features and an unnormalized alpha mask. An overall reconstruction of the ViT features is formed by linearly combining the individual reconstructions via the normalized alpha masks. We use the MLP-based decoder that is described in (Seitzer et al., 2023). Crucially, the autoencoder exclusively operates on ViT features, neither receiving RGB frames as input, nor producing them as output. Hence, the autoencoder's reconstruction loss is also measured on ViT features, providing a training signal that is more akin to perceptual similarity than similarity in RGB space. As in the experiments on the CLEVR dataset, we extract segmentations from the alpha masks. Since the alpha masks (and, correspondingly, the ViT feature maps) are of a lower resolution than the RGB frames of the MOVi-C dataset, we adopt the approach of (Seitzer et al., 2023) and bi-linearly upscale the alpha masks before computing segmentations.

**Evaluation** Following related work (Locatello et al., 2020; Seitzer et al., 2023; Kipf et al., 2022; Greff et al., 2019), we primarily judge model performance by the quality of foreground segmentations, as measured by the foreground adjusted Rand index (Rand, 1971; Hubert & Arabie, 1985) (F-ARI). Additionally, we provide figures regarding the overall segmentation performance when including the background (ARI). All models are evaluated on 1,280 scenes from the respective test sets. As reconstruction losses are rarely discussed in related work, we defer the investigation of this metric to Appendix A.

## 5.1 Object Discovery on CLEVR

In this subsection, we investigate our proposed normalization approaches on an object discovery task on the CLEVR dataset. In a first set of experiments, we follow the exact training procedure detailed in (Locatello et al., 2020) and illustrate how the different normalization methods behave as the number of slot latents $K$ is changed during inference. The effect of choosing large numbers of slot latents during training is studied in a second set of experiments. Throughout these experiments, we additionally scrutinize the impact of the object count on model performance.

**Training With 7 Slots** In this first set of experiments, we follow (Locatello et al., 2020) as closely as possible and train the previously described CNN-based architecture on the CLEVR6 dataset with 7 slots. We compare the baseline and layer normalizations proposed in (Locatello et al., 2020) to the methods discussed in Section 4. For each variant, we perform 5 training runs with different seeds. The trained models are evaluated on the CLEVR6 and CLEVR10 test sets.

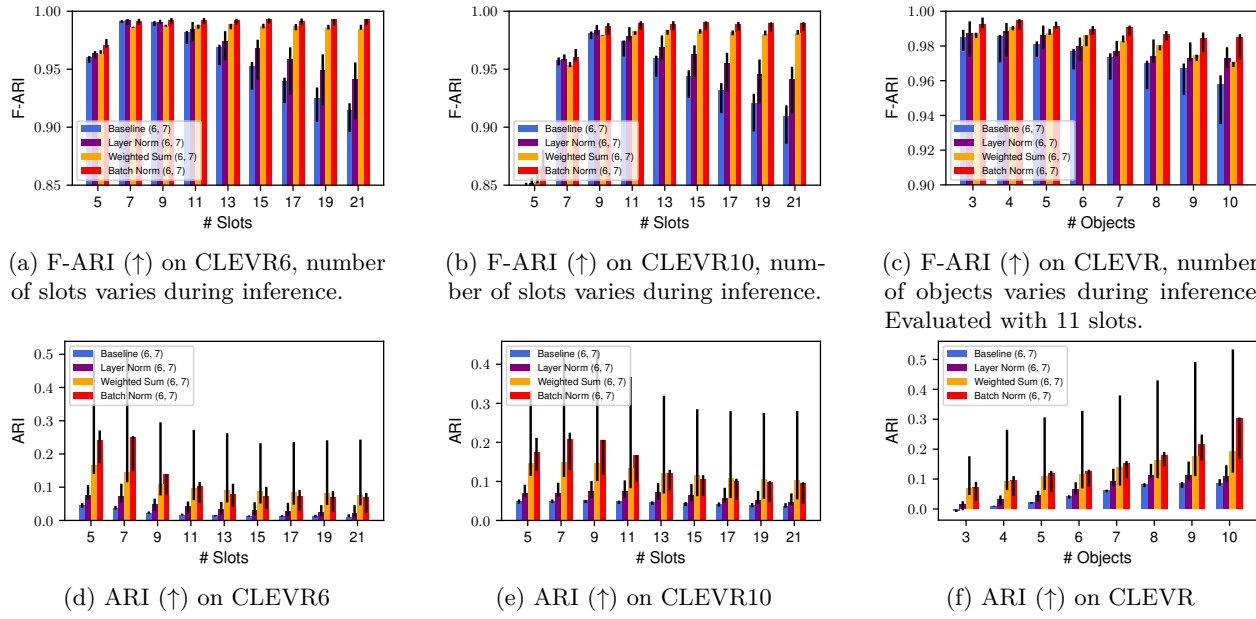

(a) F-ARI (↑) on CLEVR6, number of slots varies during inference.

(b) F-ARI (↑) on CLEVR10, number of slots varies during inference.

(c) F-ARI (↑) on CLEVR, number of objects varies during inference. Evaluated with 11 slots.

(d) ARI (↑) on CLEVR6

(e) ARI (↑) on CLEVR10

(f) ARI (↑) on CLEVR

Figure 1: Dependence of performance on slot and object count. Models are trained on CLEVR6 with 7 slots. Note the non-zero y-intercept.

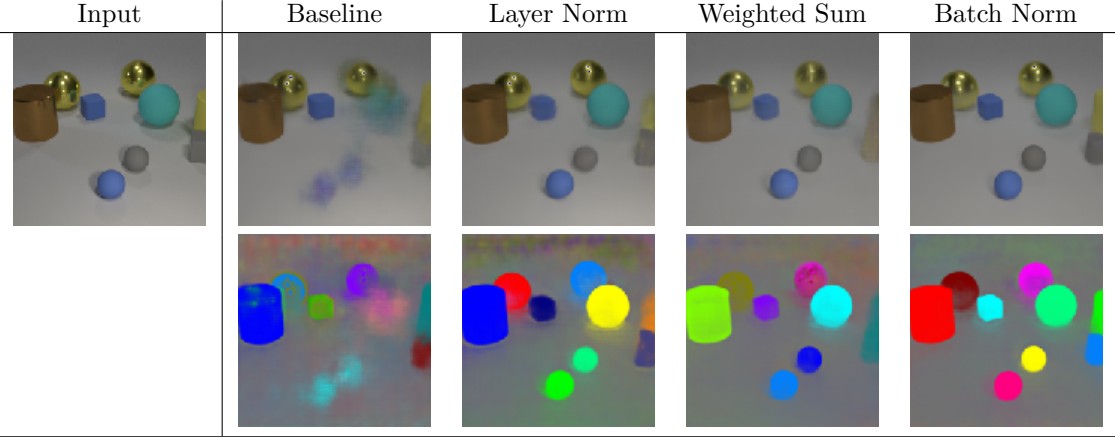

Figure 2: Qualitative results on a CLEVR10 images, showing reconstructions and (soft) segmentations. The models are trained on CLEVR6 with 7 slots and evaluated with 21 slots.

The baseline and layer norm variants lead to object-centric behavior for all 5 seeds. For the weighted sum normalization and the batch norm variant, however, we encounter two runs each in which the autoencoders decompose the input spatially instead of object-wise. Following (Locatello et al., 2020), we omit these runs in our analysis.

In Subfigures 1a and 1b, we illustrate how the foreground segmentation performance changes as we vary the number of slots during evaluation. We note that in the baseline and layer norm variants, segmentation performance deteriorates when they are presented with more than nine slots, while our proposed normalizations appear to generalize well to these changes. In particular, we observe that both of our proposed normalizations outperform the baseline when the autoencoders are evaluated with 11 slots, as is done in (Locatello et al., 2020). We show qualitative results of the different model variants at a high slot count in Figure 2 and refer to Appendix H for more visualizations.

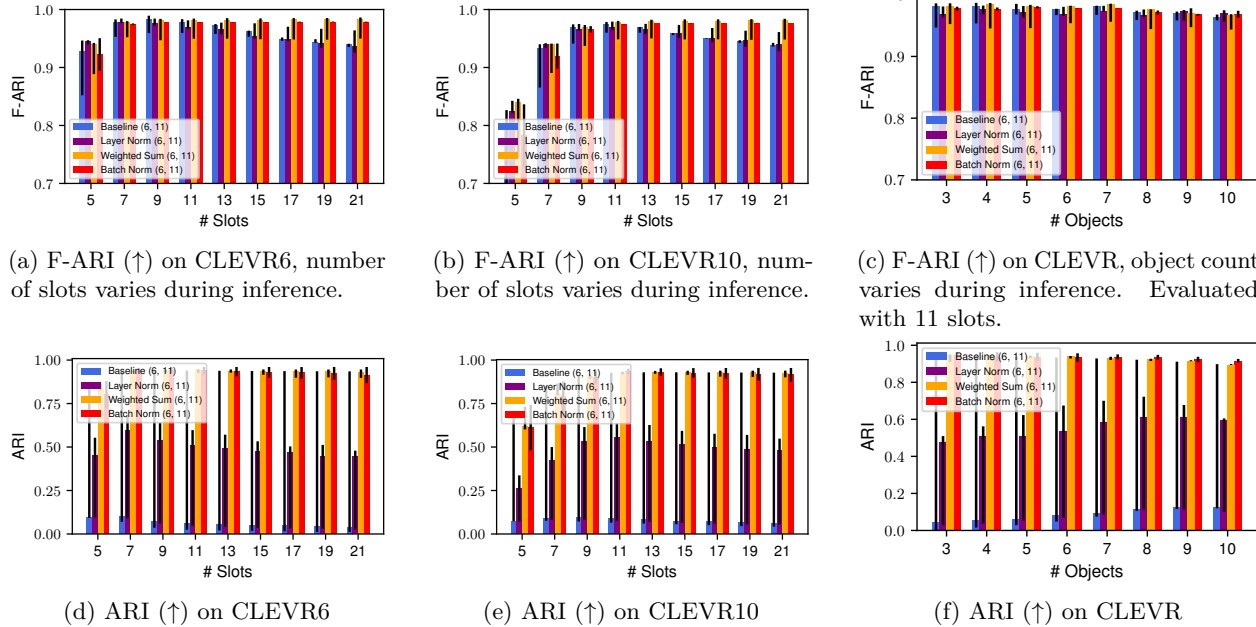

(a) F-ARI (↑) on CLEVR6, number of slots varies during inference.

(b) F-ARI (↑) on CLEVR10, number of slots varies during inference.

(c) F-ARI (↑) on CLEVR, object count varies during inference. Evaluated with 11 slots.

(d) ARI (↑) on CLEVR6

(e) ARI (↑) on CLEVR10

(f) ARI (↑) on CLEVR

Figure 3: Dependence of segmentation performance on slot and object count. Models are trained on CLEVR6 with 11 slots.

We study how performance depends on the number of objects in Subfigure 1c. Here, we evaluate each model variant with 11 slot latents on the CLEVR10 test set and plot average foreground segmentation performance dependent on the object count. Overall, we observe that our proposed normalizations outperform the baseline, independently of the object count. Also note that, across all model variants, segmentation performance trends downwards as the number of objects in the scene increases.

Subfigures 1d-1f illustrate the behavior of the overall segmentation performance when background pixels are taken into consideration. It appears that our proposed methods outperform the other two variants w.r.t. this metric, although the variablity across runs is large.

**Excess Slots During Training**  While we have so far only discussed experiments in which we increase the number of slot latents during inference, we will now outline an experiment in which the Slot Attention module is also provided with excess slots during training: Concretely, we train the autoencoders on the CLEVR6 dataset with 11 slot latents. We annotate the resulting variants with the tuple (6, 11) to underline that they were trained with 11 slots on scenes consisting of at most 6 objects. To limit computational expenses, we perform only three runs per model variant. We observe object-centric behavior in all runs for the baseline, the layer norm variant, and the weighted sum variant. For the batch normalization, we encounter one run in which the scenes are deconstructed spatially in vertical stripes. As before, we exclude this run from our analysis and are therefore left with only two runs for this variant. Considering the breadth of our other experiments and the small variability observed in this experiment, we deem this loss of information acceptable.

In this setting, the studied variants seem to perform more comparably than before w.r.t. foreground segmentation performance (Subfigures 3a-3c). Notwithstanding, we again note that the performance of the baseline and layer norm variants starts to suffer as we add additional slot latents during inference. In contrast, the performance of the two proposed variants remains more stable. In general, the foreground segmentation performance is lower than during training with few slots (Subfigures 1a-1c). All models trained with our proposed methods learn to segment the background into a single slot, leading to high overall segmentation performance (Subfigures 3d-3f). One model using the baseline normalization exhibits this behavior, while none of the models using layer normalization do so.

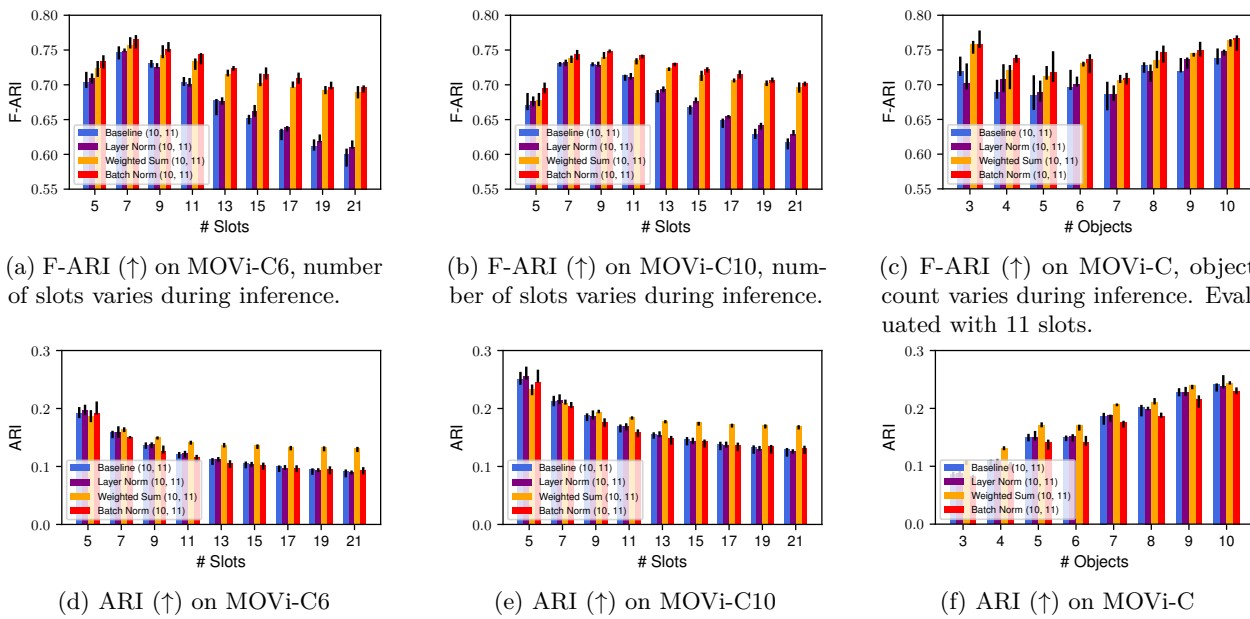

(a) F-ARI (↑) on MOVi-C6, number of slots varies during inference.

(b) F-ARI (↑) on MOVi-C10, number of slots varies during inference.

(c) F-ARI (↑) on MOVi-C, object count varies during inference. Evaluated with 11 slots.

(d) ARI (↑) on MOVi-C6

(e) ARI (↑) on MOVi-C10

(f) ARI (↑) on MOVi-C

Figure 4: Dependence of segmentation performance on slot and object count. Models are trained on MOVi-C10 with 11 slots.

## 5.2 Object Discovery on MOVi-C

To further support the validity of our proposed normalizations, we run additional experiments, using the Dinosaur (Seitzer et al., 2023) framework. As previously discussed, we train the MLP-based architecture described in (Seitzer et al., 2023) on the MOVi-C dataset. While it still is a synthetic dataset, this represents a significant step-up in complexity compared to CLEVR, approaching the complexity of real-world scenes.

**Training on MOVi-C10**   In our first set of experiments, we closely follow the setup detailed in (Seitzer et al., 2023) and train the autoencoders on the full MOVi-C10 dataset, using 11 slots. As in the previous subsection, we investigate how the foreground segmentation performance develops as we vary the number of slot latents during inference. For each model variant, we perform 5 training runs with different seeds. For sake of consistency with the other experiments, we evaluate the trained models both on the MOVi-C10 test set and on the filtered MOVi-C6 dataset.

In Subfigures 4a and 4b, we observe that our proposed normalizations generally lead to improved foreground segmentation performance compared to the baseline and layer normalization across all slot counts. In particular, both proposed normalizations outperform the baseline and layer normalization variant with 11 slots on the MOVi-C10 dataset, as can be observed in Subfigures 4a and 4b. As in our previous experiments, we note that foreground segmentation performance starts to suffer as the baseline variant is provided with excess slots during inference. While our proposed methods exhibit a similar behavior in these experiments, we note that the deterioration progresses at a slower rate. Additionally, we observe in Subfigure 4c that performance is improved across smaller object counts.

The behavior of overall segmentation performance is shown in Subfigures 4d-4f. In line with our previous observations, we note that performance suffers for all variants when excess slots are present during inference. While baseline, layer normalization and batch normalization yield comparable results w.r.t. this metric, the weighted sum variant performs noticeably better.

**Training on MOVi-C6**   While the authors of (Seitzer et al., 2023) trained autoencoders exclusively on the MOVi-C10 dataset, we will also investigate an approach that resembles the one described in (Locatello et al., 2020), and which we adopted in Subsection 5.1. Namely, we train models on the filtered MOVi-C6 dataset with 7 slots and subsequently evaluate them on both MOVi-C6 and MOVi-C10. This approach may

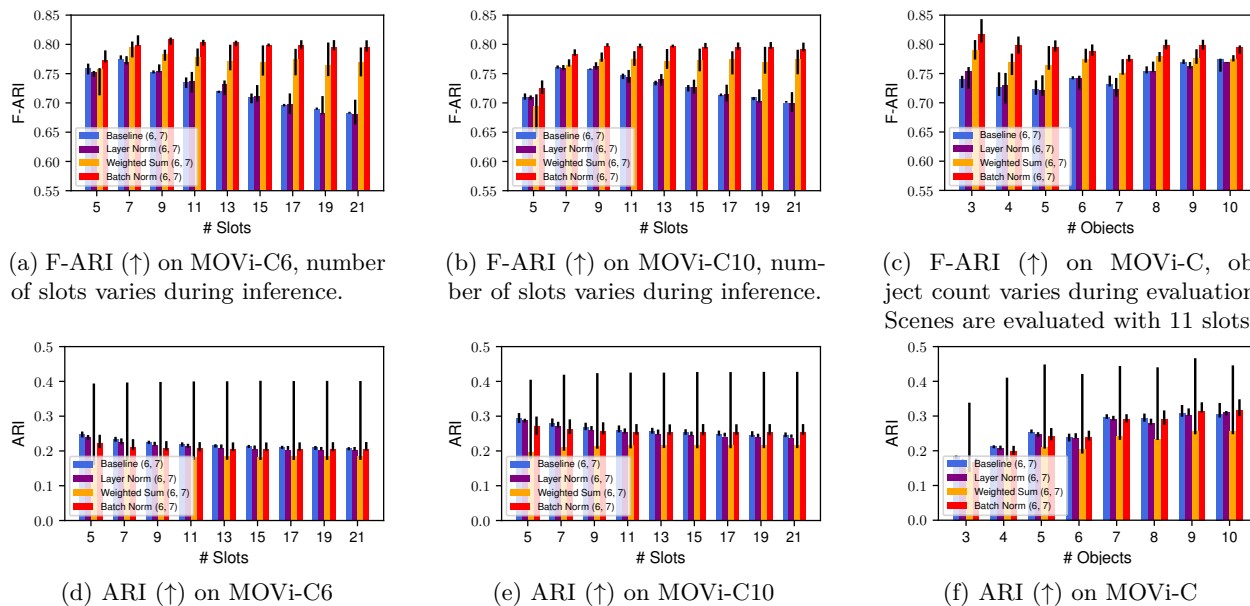

(a) F-ARI (↑) on MOVi-C6, number of slots varies during inference.

(b) F-ARI (↑) on MOVi-C10, number of slots varies during inference.

(c) F-ARI (↑) on MOVi-C, object count varies during evaluation. Scenes are evaluated with 11 slots.

(d) ARI (↑) on MOVi-C6

(e) ARI (↑) on MOVi-C10

(f) ARI (↑) on MOVi-C

Figure 5: Dependence of segmentation performance on slot and object count. Models are trained on MOVi-C6 with 7 slots.

be particularly interesting to practitioners, as reducing the number of slot latents during training can serve to greatly reduce computational effort. To limit computational expenses, we again only perform three runs per model variant.

We illustrate in Subfigures 5a-5c the behavior of the foreground segmentation performance. As in our previous experiments, we observe that both of our proposed normalizations outperform the baseline when evaluated on the MOVi-C10 test set with 11 slots. Interestingly, it can be noted that performance also improves over the models trained on the MOVi-C10 dataset with 11 slots (Figure 4). Again, we point out that excess slot latents during inference lead to a substantial deterioration of foreground segmentation performance in the baseline and layer norm models.

In Subfigures 5d-5f, we plot the behavior of overall segmentation quality. Compared to the previous experiment, varying slot count has a lesser impact on overall segmentation performance for all methods. While the differences seem unsubstantial, the baseline appears to perform best w.r.t. this metric.

**Excess Slots During Training**   In line with the experiments on the CLEVR dataset, we turn to a set of experiments that investigates the impact of excess slots during training. Similar to the corresponding setup in Subsection 5.1, we train the models on the filtered MOVi-C6 dataset, but provide them with 11 slots during training. We generally observe that both the weighted sum normalization and the batch normalization lead to improved foreground segmentations compared to the baseline and layer normalization, especially at higher slot count, as can be concluded from Subfigures 6a and 6b. Subfigure 6c additionally demonstrates once again that our proposed normalizations appear to perform at least as well as the baseline across all object counts. In Subfigures 6d-6f we find that increased slot count during inference harms overall segmentation quality. The weighted sum variant performs best, although the variablity in ARI appears large.

**Evaluation on MOVi-D**   We now investigate how the previously described models (which were trained on MOVi-C) transfer to the MOVi-D dataset. We recall that scenes of the MOVi-D dataset may contain up to 23 objects. Hence, the MOVi-D dataset allows us to study how the trained models behave when slot- and object-count are increased significantly during inference. We evaluate the models with 24 slots. In Table 1, we show the MOVi-D zero-shot performance of all 12 model variants we trained on MOVi-C. The batch norm variant performs best w.r.t. our main metric, the F-ARI. This holds true both when comparing all 12 variants to each other, and when considering the subsets of models obtained by only considering variants with any fixed

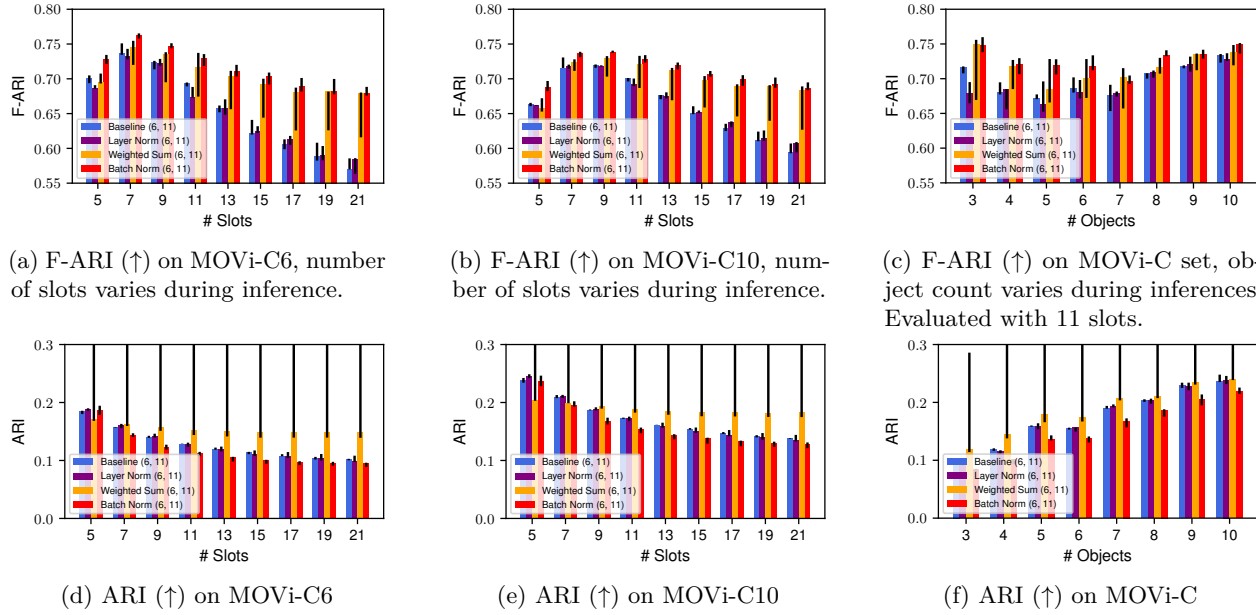

(a) F-ARI (↑) on MOVi-C6, number of slots varies during inference.

(b) F-ARI (↑) on MOVi-C10, number of slots varies during inference.

(c) F-ARI (↑) on MOVi-C set, object count varies during inferences. Evaluated with 11 slots.

(d) ARI (↑) on MOVi-C6

(e) ARI (↑) on MOVi-C10

(f) ARI (↑) on MOVi-C

Figure 6: Dependence of segmentation performance on slot and object count. Models are trained on MOVi-C6 with 11 slots.

annotation $(O, K)$. Compared to their batch normalization counterparts, the weighted sum models perform similarly w.r.t. foreground ARI, performing only slightly worse. Both the weighted sum normalization and the batch normalization produce markedly better foreground segmentations than the baseline and layer norm variants. With respect to overall segmentation quality (ARI), none of the normalization variants consistently outperform the others, which is in line with our observations on MOVi-C.

For any fixed normalization method, the $(6, 7)$ variant performs better w.r.t. F-ARI than the $(10, 11)$ and $(6, 11)$ variants. This illustrates that, firstly, training on filtered training sets with few objects can improve performance during inference on scenes with many objects; secondly, avoiding excess slots during training appears to be important, which is an observation that has been made in (Locatello et al., 2020) and which we also note when comparing Figures 1 and 3. This effect underlines the importance of strong slot-count generalization capabilities. We posit that training at low object- and slot-counts reduces the number of "unoccupied" slots during training, thereby tightening the representational slot-bottleneck, which has been postulated to encourage object-centricness (Locatello et al., 2020; Stange et al., 2023).

# 6 Related Work

In this work, we follow a longer tradition of using autoencoders to obtain semantic scene decompositions (Greff et al., 2016; 2017; 2019; Crawford & Pineau, 2019; Burgess et al., 2019; Lin et al., 2020; Engelcke et al., 2020; Locatello et al., 2020). An early work in this area is Neural-EM (Greff et al., 2017), which performs expectation maximization at the image level. It is proceeded by IODINE (Greff et al., 2019) and MONet (Burgess et al., 2019), which learn variational autoencoders. Slot Attention (Locatello et al., 2020) has already been used in many derivative works to scale object-centric learning to increasingly complex datasets. In particular, motion cues (Kipf et al., 2022; Elsayed et al., 2022; Wu et al., 2023), high-level image features (Seitzer et al., 2023) and powerful decoder models (Singh et al., 2022a;b) were found to be useful for obtaining desired behavior on real-world datasets.

Several previous works have discussed generalization capabilities of object-centric representations (Dittadi et al., 2022; Seitzer et al., 2023). That the number of slot latents has influence on model performance has been noted by Seitzer et al. (2023) where the authors illustrate that varying the number of slots has a significant impact on segmentation quality, determining whether objects are split into constituent parts or discovered in

| Variant | F-ARI ($\uparrow$) | $\ell^2$ loss ($\downarrow$) | ARI ($\uparrow$) |
|---|---|---|---|
| Baseline (10, 11) | 0.647 $\pm 0.015$ | **2.143** $\pm 0.004$ | 0.172 $\pm 0.011$ |
| Layer Norm (10, 11) | 0.660 $\pm 0.012$ | 2.146 $\pm 0.002$ | 0.171 $\pm 0.004$ |
| Weighted Sum (10, 11) | 0.722 $\pm 0.005$ | 2.223 $\pm 0.009$ | $\underline{0.204}$ $\pm 0.004$ |
| Batch Norm (10, 11) | $\underline{0.725}$ $\pm 0.006$ | 2.149 $\pm 0.004$ | 0.182 $\pm 0.011$ |
| Baseline (6, 11) | 0.640 $\pm 0.009$ | 2.220 $\pm 0.001$ | 0.176 $\pm 0.003$ |
| Layer Norm (6, 11) | 0.639 $\pm 0.011$ | 2.220 $\pm 0.004$ | 0.176 $\pm 0.005$ |
| Weighted Sum (6, 11) | 0.709 $\pm 0.044$ | 2.309 $\pm 0.014$ | $\underline{0.205}$ $\pm 0.147$ |
| Batch Norm (6, 11) | $\underline{0.711}$ $\pm 0.005$ | $\underline{2.219}$ $\pm 0.006$ | 0.175 $\pm 0.007$ |
| Baseline (6, 7) | 0.741 $\pm 0.005$ | 2.370 $\pm 0.009$ | 0.253 $\pm 0.018$ |
| Layer Norm (6, 7) | 0.742 $\pm 0.019$ | $\underline{2.365}$ $\pm 0.007$ | 0.253 $\pm 0.007$ |
| Weighted Sum (6, 7) | 0.797 $\pm 0.023$ | 2.475 $\pm 0.038$ | 0.242 $\pm 0.156$ |
| Batch Norm (6, 7) | **0.809** $\pm 0.005$ | 2.374 $\pm 0.002$ | **0.297** $\pm 0.030$ |

Table 1: Zero-shot performance on MOVi-D of models trained on MOVi-C. Evaluated with 24 slots. We show the median $\pm$ maximum deviation across multiple runs. For each metric, we underline the most advantagous variant in each section and mark the most advantagous variant across all sections in bold.

their entirety. To date, only few works investigate the role of attention normalization in the performance of Slot Attention. The first ablation evaluated for Slot Attention in the original paper (Locatello et al., 2020) is almost identical to the approach we discuss in Subsection 4.1, differing from our method crucially in that the weighted sums are not scaled by a constant, which appears to result in poor training behavior. In a second ablation, the authors modify the first ablation by normalizing the update codes via layer normalization (Ba et al., 2016). Comparable performance is reported to the original method.

Normalization in Slot Attention has been investigated in (Zhang et al., 2023), where the authors propose to use the Sinkhorn-Knopp iteration to normalize attention matrices. In contrast to our work, they do not investigate the impact on generalization capabilities and substantially increase the complexity of the SA module. More broadly, the use of the Sinkhorn-Knopp algorithm for normalization in multi-head attention modules has been scrutinized before by Sander et al. (2022).

# 7 Conclusion

Allowing models to dynamically find suitable levels of segmentation coarseness is an important problem in unsupervised object-centric representation learning. In Slot Attention it has been found that excess slots oftentimes split objects into parts or distribute responsibility for individual pixels among several slots. Hence, finding a reasonable number of slots has been crucial during training and inference. In this work, we discussed approaches for making the Slot Attention module more robust with respect to this choice. We studied normalizations of the slot-update vectors and analysed how they impact Slot Attention's ability to scale to different numbers of slots and objects during inference. On the theoretical side, we motivated this phenomenon via an analogy between Slot Attention and parameter estimation in vMF mixture models. In experiments, we demonstrated that our proposed normalization schemes increase the generalization capability of Slot Attention to varying number of slots and objects during inference. With these insights, we hope to contribute to increase performance of numerous existing and future applications of Slot Attention.

# Acknowledgement

This work was supported by Max Planck Society and Cyber Valley. The authors thank the International Max Planck Research School for Intelligent Systems (IMPRS-IS) for supporting Jan Achterhold.

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

# A    Reconstruction Loss

In this subsection, we investigate how the $\ell^2$ reconstruction loss is impacted by the choice of normalization. As in previous figures, we explore how this objective varies as object and slot-count is varied during inference. For each experiment provided in the main paper we provide a corresponding figure on the $\ell^2$ loss.

**CLEVR (6, 7)**    We first consider the experiment in which we trained a model with 7 slots on CLEVR6. We note in Subfigures 7a and 7b that all normalization approaches suffer under excess slots during inference. The layer norm variant generally performs best in this context, while the baseline and weighted sum variants perform comparably to each other. In Figure 7c, we note that reconstruction quality decreases with increasing object count for all variants. It can be observed that this deterioration progresses fastest for the baseline, slowest with batch normalization, and at a seemingly similar rate for the layer norm and weighted sum variants.

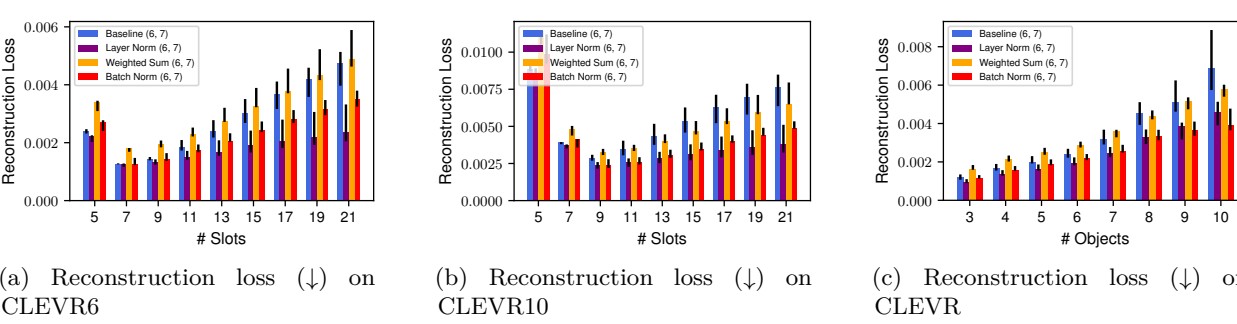

(a) Reconstruction loss (↓) on CLEVR6

(b) Reconstruction loss (↓) on CLEVR10

(c) Reconstruction loss (↓) on CLEVR

Figure 7: Dependence of reconstruction quality on slot and object count. Models are trained on CLEVR6 with 7 slots. Note the non-zero y-intercept.

**CLEVR (6, 11)**    Figure 8 illustrates the behavior of reconstruction losses for models trained on CLEVR6 with excess slots. We find that, analogous to our observations in Subsection 5.1, varying slot count during inference has a lesser effect than in the previous experiment (compare Subfigures 8a and 8b to Subfigures 7a and 7b). Generally, the weighted sum variant has the highest reconstruction loss, while the layer norm variant has the lowest. The baseline and the batch norm variant perform comparably. As before, we observe in Subfigure 8c that reconstruction loss increases with increasing object count.

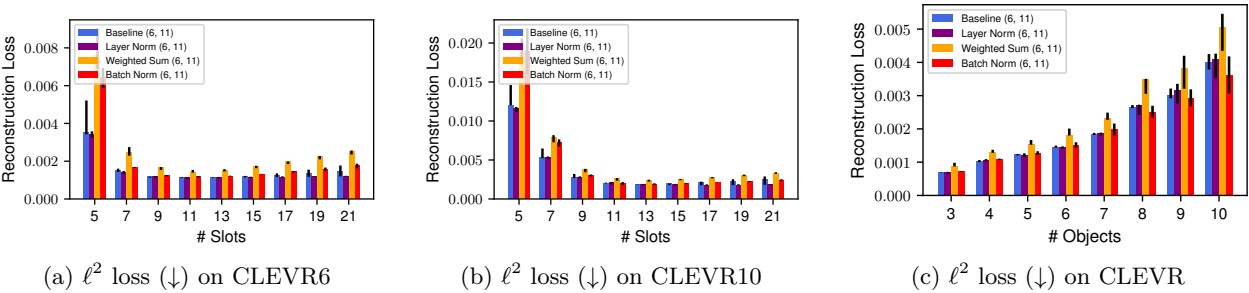

(a) $\ell^2$ loss (↓) on CLEVR6

(b) $\ell^2$ loss (↓) on CLEVR10

(c) $\ell^2$ loss (↓) on CLEVR

Figure 8: Dependence of reconstruction quality on slot and object count. Models are trained on CLEVR6 with 11 slots.

**Training on MOVi-C**    We now shift our focus to the Dinosaur models. Figure 9 shows the reconstruction loss for the models trained on MOVi-C10 with 11 slots. Contrary to previous observations, we note in Subfigures 9a and 9c that reconstruction losses improve as the slot count increases. Generally, it appears that the weighted sum variant performs worst w.r.t. the $\ell^2$ loss, while the other variants perform comparably to each other. We observe analogous behavior in Figures 10 and 11.

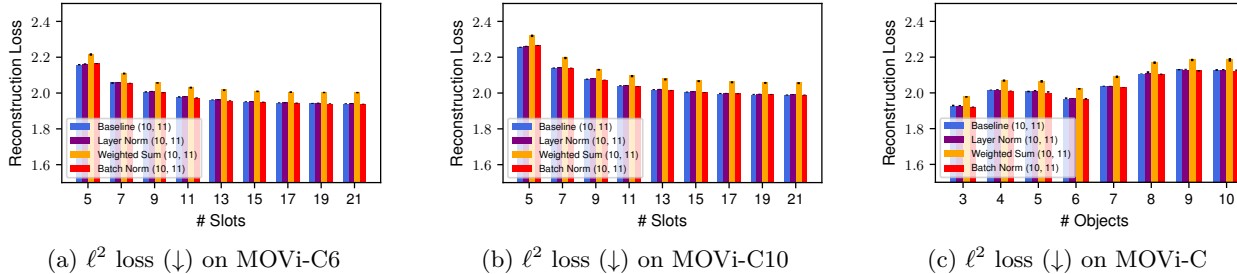

(a) $\ell^2$ loss ($\downarrow$) on MOVi-C6  (b) $\ell^2$ loss ($\downarrow$) on MOVi-C10  (c) $\ell^2$ loss ($\downarrow$) on MOVi-C

Figure 9: Dependence of reconstruction quality on slot and object count. Models are trained on MOVi-C10 with 11 slots.

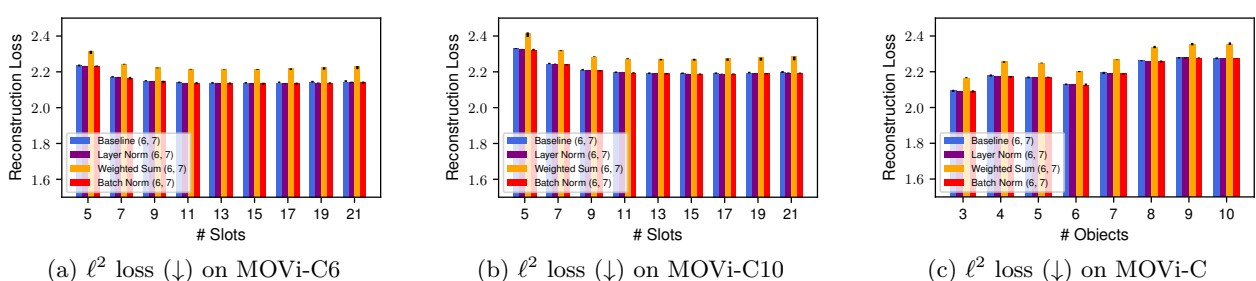

(a) $\ell^2$ loss ($\downarrow$) on MOVi-C6  (b) $\ell^2$ loss ($\downarrow$) on MOVi-C10  (c) $\ell^2$ loss ($\downarrow$) on MOVi-C

Figure 10: Dependence of reconstruction quality on slot and object count. Models are trained on MOVi-C6 with 7 slots.

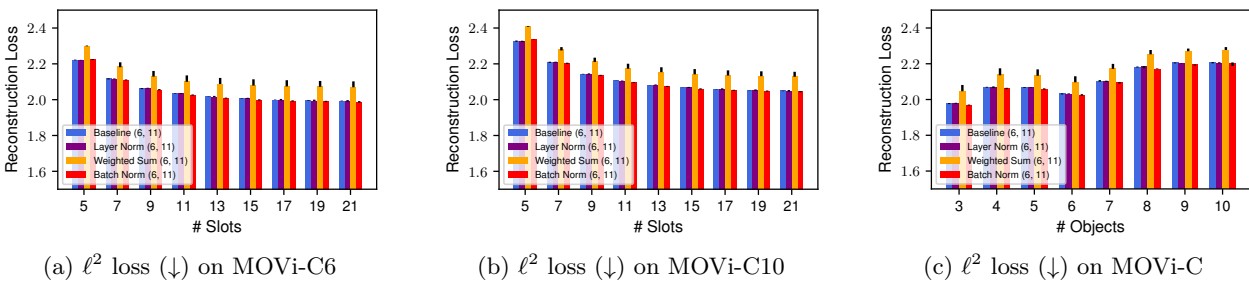

(a) $\ell^2$ loss ($\downarrow$) on MOVi-C6  (b) $\ell^2$ loss ($\downarrow$) on MOVi-C10  (c) $\ell^2$ loss ($\downarrow$) on MOVi-C

Figure 11: Dependence of reconstruction quality on slot and object count. Models are trained on MOVi-C6 with 11 slots.

## B    Lemmata on Layer Normalization

In this subsection, we will present some basic lemmata on the LayerNorm module which we use in Subsection D. For $\tilde{\boldsymbol{x}}_n \in \mathbb{R}^{D_{\text{input}}}$, LayerNorm$(\tilde{\boldsymbol{x}}_n)$ refers to the following operation:

$$\text{LayerNorm}(\tilde{\boldsymbol{x}}_n) := \text{diag}(\alpha)\frac{\tilde{\boldsymbol{x}}_n - \mathbb{E}[\tilde{\boldsymbol{x}}_n]\mathbb{1}}{\sqrt{\text{Var}[\tilde{\boldsymbol{x}}_n] + \epsilon}} + \beta \tag{14}$$

where $\alpha, \beta \in \mathbb{R}^D$ are learnable parameters, $\epsilon > 0$ is a constant, $\mathbb{1}$ is the all-ones vector, and $\mathbb{E}[\tilde{\boldsymbol{x}}_k]$ and $\text{Var}[\tilde{\boldsymbol{x}}_k]$ are the expectation and variance of the entries of $\tilde{\boldsymbol{x}}_k$, respectively:

$$\mathbb{E}[\tilde{\boldsymbol{x}}_n] := \frac{1}{D}\tilde{\boldsymbol{x}}_n^\top \mathbb{1} \tag{15} \qquad\qquad \text{Var}[\tilde{\boldsymbol{x}}_n] := \frac{1}{D}\|\tilde{\boldsymbol{x}}_n - \mathbb{E}[\tilde{\boldsymbol{x}}_n]\mathbb{1}\|_2^2 \tag{16}$$

Given these definitions, we present the following lemma, giving an expression for the affine hull of the image of the layer normalization module.

**Lemma 1.** *The vector $\tilde{\boldsymbol{x}}_k - \mathbb{E}[\tilde{\boldsymbol{x}}_k]\mathbb{1}$ is the orthogonal projection of $\tilde{\boldsymbol{x}}_k$ onto the orthogonal complement of the span of the all-ones vector $\langle \mathbb{1} \rangle^\perp \subsetneq \mathbb{R}^{D_{input}}$. Hence, the image of LayerNorm is contained in a $(D_{input} - 1)$-dimensional affine subspace of $\mathbb{R}^{D_{input}}$. More concretely, the affine hull of the image of a LayerNorm module with parameters $\alpha, \beta \in \mathbb{R}^{D_{input}}$ is given via:*

$$\text{aff}\left(\text{LayerNorm}\left[\mathbb{R}^{D_{input}}\right]\right) = \text{diag}(\alpha)\mathbb{1}^\perp + \beta \tag{17}$$

*Here and in the following, the left-multiplication of the diagonal matrix $\text{diag}(\alpha)$ with any set (e.g. the vector space $\mathbb{1}^\perp$) refers to the image of that set under the left-multiplication.*

*Proof.* Define $e_1 := 1/\sqrt{D}\mathbb{1}$ and extend to an orthonormal basis $e_1, ..., e_{D_{\text{input}}}$ via Gram-Schmidt. Then we have $\langle \mathbb{1} \rangle^\perp = \langle e_2, ..., e_{D_{\text{input}}} \rangle$ and for an arbitrary $\tilde{\boldsymbol{x}}_n$ we have the orthogonal decomposition:

$$\tilde{\boldsymbol{x}}_n = \sum_{d=1}^{D_{\text{input}}} \langle \tilde{\boldsymbol{x}}_n, e_d \rangle e_d \tag{18}$$

The orthogonal projection onto $\langle \mathbb{1} \rangle^\perp$ is then given by

$$\sum_{d=2}^{D_{\text{input}}} \langle \tilde{\boldsymbol{x}}_n, e_d \rangle e_d = \tilde{\boldsymbol{x}}_n - \langle \tilde{\boldsymbol{x}}_n, e_1 \rangle e_1 \tag{19}$$

We note that we may rewrite

$$\langle \tilde{\boldsymbol{x}}_n, e_1 \rangle e_1 = \left(\frac{1}{\sqrt{D}}\tilde{\boldsymbol{x}}_n^\top \mathbb{1}\right)\frac{1}{\sqrt{D}}\mathbb{1} = \left(\frac{1}{D}\tilde{\boldsymbol{x}}_n^\top \mathbb{1}\right)\mathbb{1} = \mathbb{E}[\tilde{\boldsymbol{x}}_k]\mathbb{1} \tag{20}$$

Hence, $\tilde{\boldsymbol{x}}_n - \mathbb{E}[\tilde{\boldsymbol{x}}_n]\mathbb{1}$ indeed realizes the orthogonal projection of $\tilde{\boldsymbol{x}}_n$ onto $\langle \mathbb{1} \rangle^\perp$. Clearly, $\langle \mathbb{1} \rangle^\perp$ is a $(D_{\text{input}} - 1)$-dimensional linear subspace of $\mathbb{R}^{D_{\text{input}}}$ (being the span of $D_{\text{input}} - 1$ many vectors). We note that the affine hull commutes with affine functions. Hence, we have:

$$\text{aff}\left(\text{LayerNorm}\left[\mathbb{R}^{D_{\text{input}}}\right]\right) = \text{diag}(\alpha)\,\text{aff}\left(g\left[\mathbb{R}^{D_{\text{input}}}\right]\right) + \beta \tag{21}$$

where the function $g$ is given as:

$$g(\tilde{\boldsymbol{x}}_n) := \frac{\tilde{\boldsymbol{x}}_n - \mathbb{E}[\tilde{\boldsymbol{x}}_n]\mathbb{1}}{\sqrt{\text{Var}[\tilde{\boldsymbol{x}}_n] + \epsilon}} \tag{22}$$

We have already shown that $\tilde{\boldsymbol{x}}_k - \mathbb{E}[\tilde{\boldsymbol{x}}_k]\mathbb{1}$ is exactly the orthogonal projection onto $\mathbb{1}^\perp$. Since $g$ differs from this projection via a multiplicative factor, the affine span of its image is exactly the affine span of the projection, which is $\mathbb{1}^\perp$. Thus, we have shown:

$$\text{aff}\left(\text{LayerNorm}\left[\mathbb{R}^{D_{\text{input}}}\right]\right) = \text{diag}(\alpha)\mathbb{1}^\perp + \beta \tag{23}$$

$\square$

We can write these affine subspaces in a particular fashion, as we show in the following lemma. This result will be useful for explicitly writing down a function satisfying Proposition 2.

**Lemma 2.** *For any affine subspace $A \subset \mathbb{R}^{D_{input}}$, there exists a unique vector $\boldsymbol{a} \in A$ and a unique vectorspace $V \subset \mathbb{R}^{D_{input}}$ such that $\boldsymbol{a} \in V^\perp$ is orthogonal to $V$ and we have $A = \boldsymbol{a} + V$.*

*Proof.* Given any affine subspace $A \subset \mathbb{R}^{D_{\text{input}}}$, let $\boldsymbol{a} \in A$ be the orthogonal projection of the origin onto $A$. By definition of the orthogonal projection, for any other $\boldsymbol{a}' \in A$, we have $(\boldsymbol{a} - 0)^\top (\boldsymbol{a}' - \boldsymbol{a}) = 0$. Moreover, $V := A - \boldsymbol{a}$ is a linear subspace of $\mathbb{R}^{D_{\text{input}}}$ with $\boldsymbol{a} + V = A$. For any arbitrary $\boldsymbol{v} \in V$, we may write $\boldsymbol{v} = \boldsymbol{a}' - \boldsymbol{a}$ for some $\boldsymbol{a}' \in A$ and we have

$$\boldsymbol{a}^\top \boldsymbol{v} = (\boldsymbol{a} - 0)^\top (\boldsymbol{a}' - \boldsymbol{a}) = 0 \tag{24}$$

Hence, $\boldsymbol{a} \in V^\perp$ holds. We have so far shown the existence of some vector space $V$ and $\boldsymbol{a} \in V^\perp$ with $A = \boldsymbol{a} + V$.

We now show that this decomposition $A = \boldsymbol{a} + V$ is unique. Assume that there exists another vector space $V' \subset \mathbb{R}^{D_{\text{input}}}$ and $\boldsymbol{a}' \in V'^\perp$ with $\boldsymbol{a} + V = \boldsymbol{a}' + V'$. We will show that $V = V'$ and $\boldsymbol{a} = \boldsymbol{a}'$ must hold. Rearranging, we find $V' = (\boldsymbol{a} - \boldsymbol{a}') + V$ and $V = V' - (\boldsymbol{a} - \boldsymbol{a}')$. From the first equation, we may conclude $\boldsymbol{a} - \boldsymbol{a}' \in V'$. Since $\boldsymbol{a} - \boldsymbol{a}'$ is a vector in $V'$, which is closed under subtraction, we find $V' - (\boldsymbol{a} - \boldsymbol{a}') = V'$. Hence, substituting into the previous equation, we conclude $V = V'$. We now show that we must also have $\boldsymbol{a} = \boldsymbol{a}'$. We compute:

$$(\boldsymbol{a} - \boldsymbol{a}')^\top (\boldsymbol{a} - \boldsymbol{a}') = \boldsymbol{a}^\top (\boldsymbol{a} - \boldsymbol{a}') - \boldsymbol{a}'^\top (\boldsymbol{a} - \boldsymbol{a}') \tag{25}$$

and recall that $\boldsymbol{a} - \boldsymbol{a}' \in V$ holds. By assumption, we have $\boldsymbol{a}, \boldsymbol{a}' \in V^\perp$ and therefore $\boldsymbol{a}^\top (\boldsymbol{a} - \boldsymbol{a}') = 0$ and $\boldsymbol{a}'^\top (\boldsymbol{a} - \boldsymbol{a}') = 0$. Hence, $(\boldsymbol{a} - \boldsymbol{a}')^\top (\boldsymbol{a} - \boldsymbol{a}') = 0$ and we conclude from the positive-definiteles of the scalar product that $\boldsymbol{a} = \boldsymbol{a}'$ does indeed hold. $\square$

Finally, we show that for almost all parameters of the value map $v$ and the layer normalization module, the translation vector $\boldsymbol{a}$ from the previous lemma does not vanish. Again, this will be crucial to show that a function satisfying Proposition 2 exists almost always.

**Lemma 3.** *Denote $D := D_{input}$ and let $v : \mathbb{R}^D \to \mathbb{R}^D$ be a linear map that is parametrized via a matrix $B \in \mathbb{R}^{D \times D}$ which acts, say, via left-multiplication. Consider parameters $\alpha, \beta \in \mathbb{R}^D$ of the layer normalization. Consider the affine hull of the image of the composition of $v$ and the layer normalization:*

$$A := \text{aff}\left( (v \circ \text{LayerNorm})\left[\mathbb{R}^D\right] \right) \tag{26}$$

*As detailed in Lemma 2, we may write $A$ uniquely as $\boldsymbol{a} + V$, where $\boldsymbol{a}$ and $V$ depend on the parameters of $v$ and the layer normalization. In the following, we abuse notation by not making this dependence explicit. For almost all parameters $(B, \alpha, \beta)$ (w.r.t. the Lebesgue measure on $\mathbb{R}^{D \times D} \times \mathbb{R}^D \times \mathbb{R}^D$), we have $\boldsymbol{a} \neq 0$.*

*Proof.* Let the set

$$U := \{(B, \alpha, \beta) \in \mathbb{R}^{D \times D} \times \mathbb{R}^D \times \mathbb{R}^D \mid \boldsymbol{a} = 0\} \tag{27}$$

represent the parameters for which $\boldsymbol{a}$ vanishes. We will show that this is a nullset w.r.t. Lebesgue measure. It is a standard fact that the set of non-invertible matrices $\text{GL}_D(\mathbb{R})^C \subset \mathbb{R}^{D \times D}$ is a Lebesgue-nullset. Hence, it is already sufficient to prove that the set

$$U' := U \cap (\text{GL}_D(\mathbb{R}) \times \mathbb{R}^D \times \mathbb{R}^D) \tag{28}$$

is a nullset. As we saw from the proof of Lemma 2, the vector $\boldsymbol{a}$ is exactly the orthogonal projection of the origin onto $A$. Hence, $\boldsymbol{a} = 0$ holds iff $0 \in A$ holds. Recalling Lemma 1, we know

$$\begin{aligned} A &:= \text{aff}\left( (v \circ \text{LayerNorm})\left[\mathbb{R}^D\right] \right) = v(\text{aff LayerNorm}[\mathbb{R}^D]) \\ &= B(\text{diag}(\alpha)\mathbb{1}^\perp + \beta) \end{aligned} \tag{29}$$

As we may assume that $B$ is invertible, it has a trivial kernel. Hence, we have $0 \in A$ iff $0 \in \mathrm{diag}(\alpha)\mathbb{1}^{\perp} + \beta$. From this requirement, we obtain an explicit expression for the set $U'$:

$$
\begin{aligned}
U' &= \mathrm{GL}_D(\mathbb{R}) \times \{(\alpha, \beta) \mid \alpha \in \mathbb{R}^D, \, \beta \in -\mathrm{diag}(\alpha)\mathbb{1}^{\perp}\} \\
&= \mathrm{GL}_D(\mathbb{R}) \times \{(\alpha, -\mathrm{diag}(\alpha)u) \mid \alpha \in \mathbb{R}^D, \, u \in \mathbb{1}^{\perp}\}
\end{aligned}
\tag{30}
$$

We conclude by noting that $\{(\alpha, -\mathrm{diag}(\alpha)u) \mid \alpha \in \mathbb{R}^D, \, u \in \mathbb{1}^{\perp}\}$ is a nullset in $\mathbb{R}^D \times \mathbb{R}^D$, as it is the image of a nullset under a continuously differentiable function. $\square$

## C  Proof of Proposition 1

*Proof.* Assume by contradiction that such a function $f : \mathbb{R}^D \to \mathbb{R}$ exists. Consider now the setting in which we have $K := 1$ and $\tilde{\boldsymbol{\theta}}_1 := 0$. Denote the resulting attention matrix by $\boldsymbol{\Gamma}^{(1)}$ and the resulting update code $\boldsymbol{u}_1$ by $\boldsymbol{w}_1^{(1)}$. Consider also the setting in which we have $K := 2$ and $\tilde{\boldsymbol{\theta}}_1 = \tilde{\boldsymbol{\theta}}_2 := 0$. Denote the resulting attention matrix by $\boldsymbol{\Gamma}^{(2)}$ and the resulting update codes $\boldsymbol{u}_1, \boldsymbol{u}_2$ by $\boldsymbol{w}_1^{(2)}, \boldsymbol{w}_2^{(2)}$. One may now easily verify that all entries of $\boldsymbol{\Gamma}^{(1)}$ are 1 and that all entries of $\boldsymbol{\Gamma}^{(2)}$ are $1/2$. Hence, we have

$$
w_1^{(1)} = \frac{\sum_{n=1}^{N} \gamma_{n,1}^{(1)} \boldsymbol{v}_n}{\sum_{n=1}^{N} \gamma_{n,1}^{(1)}} = \frac{\sum_{n=1}^{N} \boldsymbol{v}_n}{N}
\tag{31}
$$

and

$$
w_1^{(2)} = w_2^{(2)} = \frac{\sum_{n=1}^{N} \frac{1}{2} \boldsymbol{v}_n}{\sum_{n=1}^{N} \frac{1}{2}} = \frac{\sum_{n=1}^{N} \boldsymbol{v}_n}{N}
\tag{32}
$$

By equation (32) and our assumption (namely, that equation (9) holds), we have:

$$
f\left(\sum_{n=1}^{N} \boldsymbol{v}_n/N\right) = f\left(w_1^{(2)}\right) = \frac{\sum_{n=1}^{N} \gamma_{n,k}^{(2)}}{N} = \frac{1}{2}
\tag{33}
$$

At the same time, we also deduce from equation (31):

$$
f\left(\sum_{n=1}^{N} \boldsymbol{v}_n/N\right) = f\left(w_1^{(1)}\right) = \frac{\sum_{n=1}^{N} \gamma_{n,k}^{(1)}}{N} = 1
\tag{34}
$$

This contradicts equation (33). $\square$

## D  Proof of Proposition 2

*Proof.* We recall from Lemma 3 that for almost all parameters of the value map $v$ and the layer normalization module, we may choose a vector $\boldsymbol{a} \in \mathbb{R}^D \setminus \{0\}$ and a vector space $W \subset \mathbb{R}^D$ such that $\boldsymbol{a} \in W^{\perp}$ holds and for any input $\tilde{\boldsymbol{x}}_n$, the corresponding values $\boldsymbol{v}_n$ lie in $\boldsymbol{a} + W$. Given fixed parameters (that do not lie in the nullset outlined in Lemma 3), fix such a vector $\boldsymbol{a}$.

Now, we define the function $f : \mathbb{R}^D \to \mathbb{R}$ via:

$$
f(\boldsymbol{u}) := C \frac{\boldsymbol{a}^{\top} \boldsymbol{u}}{N \|\boldsymbol{a}\|_2^2}
\tag{35}
$$

The values $\boldsymbol{v}_n$ may be written as

$$
\boldsymbol{v}_n = \boldsymbol{a} + \boldsymbol{w}_n
\tag{36}
$$

where $\boldsymbol{w}_n \in W$ and $\boldsymbol{a}^\top \boldsymbol{w}_n = 0$ holds. Hence, recalling that we define $\boldsymbol{u}_k := \frac{1}{C} \sum_{n=1}^{N} \gamma_{n,k} \boldsymbol{v}_n$, we may now compute:

$$
\begin{aligned}
f(\boldsymbol{u}_k) &= C \frac{\boldsymbol{a}^\top}{N \|\boldsymbol{a}\|_2^2} \frac{1}{C} \sum_{n=1}^{N} \gamma_{n,k} (\boldsymbol{a} + \boldsymbol{w}_n) = \frac{1}{N \|\boldsymbol{a}\|_2^2} \sum_{n=1}^{N} \gamma_{n,k} \boldsymbol{a}^\top (\boldsymbol{a} + \boldsymbol{w}_n) \\
&= \frac{1}{N \|\boldsymbol{a}\|_2^2} \sum_{n=1}^{N} \gamma_{n,k} \boldsymbol{a}^\top \boldsymbol{a} \\
&= \frac{\sum_{n=1}^{N} \gamma_{n,k}}{N}
\end{aligned}
\tag{37}
$$

which is what we wanted to show. $\qquad\square$

## E   Technical Details Regarding Object Discovery on CLEVR

While we closely follow the descriptions of (Locatello et al., 2020), we use a *re-implementation* of the method in our experiments.

**Data**   As in (Locatello et al., 2020), we use the extended CLEVR dataset that is provided in (Kabra et al., 2019). This version of the dataset consists of 100,000 images in total, each being of dimension $320 \times 240$. We follow (Locatello et al., 2020; Greff et al., 2019; Burgess et al., 2019) in the pre-processing of this data: We use 70,000 images for training and hold out 15,000 for validation and testing. We perform a square center crop of size 192 to increase the space occupied by objects. The cropped images are then bilinearly scaled to shape $128 \times 128$. The corresponding ground-truth segmentation masks are pre-processed analogously, using nearest-neighbor interpolation in place of bi-linear interpolation. In contrast to (Locatello et al., 2020), we augment the data by performing random horizontal flips. Before feeding it to the autoencoder, the RGB data is scaled to the interval $[-1, 1]$.

**Architecture**   We follow the autoencoder architecture described in (Locatello et al., 2020) as closely as possible. Conceptually, we divide the autoencoder into three distinct entities: The *encoder* processes the input image and produces a set of tokens. The *Slot Attention module* processes these tokens and yields a latent slot representation. The *decoder* decodes the latent representation into a reconstruction of the input.

The encoder consists of a convolutional network, which we describe (as in (Locatello et al., 2020)) in Table 2.

| Type | In Shape | Out Shape | Activation | Comment |
|------|----------|-----------|------------|---------|
| Conv $5 \times 5$ | $128 \times 128 \times 3$ | $128 \times 128 \times 64$ | ReLU | stride:1 |
| Conv $5 \times 5$ | $128 \times 128 \times 64$ | $128 \times 128 \times 64$ | ReLU | stride:1 |
| Conv $5 \times 5$ | $128 \times 128 \times 64$ | $128 \times 128 \times 64$ | ReLU | stride:1 |
| Conv $5 \times 5$ | $128 \times 128 \times 64$ | $128 \times 128 \times 64$ | ReLU | stride:1 |
| Pos. Embed | $128 \times 128 \times 64$ | $128 \times 128 \times 64$ | - | - |
| Spatially Flatten | $128 \times 128 \times 64$ | $(128 \cdot 128) \times 64$ | - | - |
| Layer Norm | $(128 \cdot 128) \times 64$ | $(128 \cdot 128) \times 64$ | - | per 64-dim token |
| Affine | $(128 \cdot 128) \times 64$ | $(128 \cdot 128) \times 64$ | ReLU | per 64-dim token |
| Affine | $(128 \cdot 128) \times 64$ | $(128 \cdot 128) \times 64$ | - | per 64-dim token |

Table 2: Encoder network for experiments on CLEVR

We implement the positional embedding as in (Locatello et al., 2020). Namely, to positionally embed a tensor $X$ of shape $W \times H \times C$, we construct a $W \times H \times 4$ tensor $P$ in which each of the four channels is a linear ramp spanning between 0 and 1, either progressing horizontally or vertically and in either of the two possible

directions (e.g. left or right). In order to embed the feature $X_{i,j,:}$, we learn an affine layer $\mathbb{R}^4 \to \mathbb{R}^C$ and compute the embedded feature:

$$X_{i,j,:} + \texttt{affine}(P_{i,j,:}) \tag{38}$$

The resulting set of input tokens is then processed by the Slot Attention module, which we implement as described in (Locatello et al., 2020). The key, query, and value maps $q$, $k$, $v$ use the common dimension $D = 64$. Slots are also 64-dimensional. The residual MLP that is used to update the slot latents has a single hidden layer of size 128.

We decode each slot separately, using a spatial broadcast decoder. Once again, we closely follow the approach of (Locatello et al., 2020) and detail the decoder architecture in Table 3.

| Type | In Shape | Out Shape | Activation | Comment |
|---|---|---|---|---|
| Spatial Broadcast | 64 | $8 \times 8 \times 64$ | - | for single slot |
| Pos. Embed | $8 \times 8 \times 64$ | $8 \times 8 \times 64$ | - | - |
| Transposed Conv $5 \times 5$ | $8 \times 8 \times 64$ | $16 \times 16 \times 64$ | ReLU | stride:2 padding:2 out padding: 1 |
| Transposed Conv $5 \times 5$ | $16 \times 16 \times 64$ | $32 \times 32 \times 64$ | ReLU | as above |
| Transposed Conv $5 \times 5$ | $32 \times 32 \times 64$ | $64 \times 64 \times 64$ | ReLU | as above |
| Transposed Conv $5 \times 5$ | $64 \times 64 \times 64$ | $128 \times 128 \times 64$ | ReLU | as above |
| Transposed Conv $5 \times 5$ | $128 \times 128 \times 64$ | $128 \times 128 \times 64$ | ReLU | stride:1 padding:2 |
| Transposed Conv $3 \times 3$ | $128 \times 128 \times 64$ | $128 \times 128 \times 4$ | - | stride:1 padding:1 |

Table 3: Decoder network for experiments on CLEVR

The 4 channels of the output of the decoder are split into RGB channels and an unnormalized alpha channel. The alpha channels are normalized via a softmax operation across all slots, and the RGB reconstructions are blended to produce an entire reconstruction.

**Training** We closely follow the training procedure of (Locatello et al., 2020). Namely, we train the autoencoder with an $\ell^2$ reconstruction loss, utilize 3 Slot Attention iterations during training, and use an Adam (Kingma & Ba, 2015) optimizer. The models are trained for 500,000 steps. As the authors of (Locatello et al., 2020), we linearly warm up the learning rate over the course of the first 10,000 steps, after which it attains a peak value of $4 \cdot (0.5)^{0.1} \cdot 10^{-4}$. Subsequently, we decay it over the course of the remaining steps, with a half life of 100,000 steps. We use a batch size of 64.

**Evaluation** During evaluation, we use 5 Slot Attention iterations.

## F   Technical Details Regarding Object Discovery on MOVi-C

While we closely follow the descriptions of (Seitzer et al., 2023), we use a *re-implementation* of the method in our experiments.

**Data** We use the MOVi-C dataset from (Greff et al., 2022). In total, it contains 10,986 video sequences, each consisting of 24 frames. We hold out 250 of the sequence for validation and 999 for testing. Following (Seitzer et al., 2023), we pre-process the frames from the dataset by bicubicly resizing them to shape $224 \times 224$. We use a pretrained vision transformer to pre-compute image features from these resized frames. Specifically, we use the model `vit_base_patch8_224_dino` from the timm (Wightman, 2019) repository. After dropping the class token, this model provides a $28 \times 28 \times 768$ feature map for each frame. We save these feature maps at

half precision (16 bit floating point) and use them during training. Alongside, we save the resized frames and analogously resized (via nearest-neighbor interpolation) ground truth segmentations.

**Architecture**   We closely follow the MLP-based autoencoder architecture detailed in (Seitzer et al., 2023). The goal of this autoencoder is to encode and decode the pre-computed ViT features. Importantly, it does not operate on RGB images. As before, we conceptually divide the autoencoder into the *encoder*, the *Slot Attention module*, and the *decoder*.

The encoder processes each ViT feature separately via a shared MLP. In contrast to the encoder we used in the experiments on the CLEVR dataset, no positional embedding is employed, as the ViT features still contain a sufficient amount of positional information (see the discussions in (Seitzer et al., 2023)). We provide a detailed description of this architecture in Table 4.

| Type | In Shape | Out Shape | Activation | Comment |
|------|----------|-----------|------------|---------|
| Layer Norm | $28 \times 28 \times 768$ | $28 \times 28 \times 768$ | - | per feature |
| Affine | $28 \times 28 \times 768$ | $28 \times 28 \times 768$ | ReLU | per feature |
| Affine | $28 \times 28 \times 768$ | $28 \times 28 \times 128$ | - | per feature |

Table 4: Encoder network for experiments on MOVi-C

The set of tokens that is produced by the encoder is processed by the Slot Attention module to produce a latent slot representation. Slots, keys, values, and queries are 128-dimensional. The residual MLP that is used to update the slot latents has a single hidden layer of dimension 512.

In order to produce a reconstruction of the ViT features, an MLP-based decoder architecture is used. Conceptually, the decoder resembles the one we used in the experiments on the CLEVR dataset: Each slot is decoded separately into a partial reconstruction and an unnormalized alpha channel. A complete reconstruction is formed by normalizing the alpha channels across slots and blending the partial reconstructions. Each slot is broadcasted spatially before decoding and a positional embedding is added. Once again, the decoder consists of an MLP that operates on each spatial feature separately. We provide a detailed description of the architecture in Table 5

| Type | In Shape | Out Shape | Activation | Comment |
|------|----------|-----------|------------|---------|
| Spatial Broadcast | $128$ | $28 \times 28 \times 128$ | - | - |
| Pos. Embedding | $28 \times 28 \times 128$ | $28 \times 28 \times 128$ | - | - |
| Affine | $28 \times 28 \times 128$ | $28 \times 28 \times 1024$ | ReLU | per feature |
| Affine | $28 \times 28 \times 1024$ | $28 \times 28 \times 1024$ | ReLU | per feature |
| Affine | $28 \times 28 \times 1024$ | $28 \times 28 \times 1024$ | ReLU | per feature |
| Affine | $28 \times 28 \times 1024$ | $28 \times 28 \times (768 + 1)$ | - | per feature |

Table 5: Decoder network for experiments on MOVi-C

The positional embedding used in Table 5 differs significantly from the one we used in the experiments on CLEVR. Namely, given an input tensor $X$ of shape $W \times H \times C$, we positionally embed the feature $X_{i,j,:}$ by learning a tensor $P$ of shape $W \times H \times C$ and computing:

$$X_{i,j,:} + P_{i,j,:} \tag{39}$$

**Training**   We follow the training procedure detailed in (Seitzer et al., 2023). Namely, we train the autoencoder via an $\ell^2$ reconstruction loss and use 3 Slot Attention iterations during training. The batch size is 64 and we employ an Adam optimizer. As before, a learning rate schedule consisting of a linear increase and exponential decay is used. Here, the peak learning rate is $4 \cdot 10^{-4}$, which is reached after 10,000 steps. Afterwards, the learning rate decays with a half life of 100,000 steps. The models are trained for 500,000 steps in total.

**Evaluation**   During evaluation, we also utilize 3 Slot Attention iterations. Since the alpha masks that are produced by our model are of shape $28 \times 28$, we cannot directly compare them to ground truth segmentations, which are of shape $224 \times 224$. We follow the approach of (Seitzer et al., 2023) and bi-linearly upscale the alpha masks to shape $224 \times 224$ before deriving segmentations, which we then compare to the ground-truth.

## G   Pseudocode

In Algorithms 1 and 2, we illustrate how the weighted sum and batch norm variants differ from the weighted mean variant in pseudo PyTorch code. We illustrate this in a diff format.

---

**Algorithm 1** Diff of Weighted Sum Variant

```
1        ...
2        bs, N, d_in = inputs.shape
3        k, v = self.key_map(inputs), self.value_map(inputs)
4        for idx in range(num_iters):
5            slots_prev = slots
6            slots = self.norm_slots(slots)
7            q = self.query_map(slots)
8            dots = torch.einsum("bid,bjd->bij", q, k) / np.sqrt(q.size(-1))
9            attn = dots.softmax(dim=1)
10 -         attn = (attn + eps) / (attn + eps).sum(dim=-1, keepdim=True)
11           updates = torch.einsum("bjd,bij->bid", v, attn)
12 +         updates = updates / N
13           ...
```

---

**Algorithm 2** Diff of Batch Norm Variant

```
1        ...
2        bs, N, d_in = inputs.shape
3        k, v = self.key_map(inputs), self.value_map(inputs)
4  +     var, mean = None, None
5        for idx in range(num_iters):
6            slots_prev = slots
7            slots = self.norm_slots(slots)
8            q = self.query_map(slots)
9            dots = torch.einsum("bid,bjd->bij", q, k) / np.sqrt(q.size(-1))
10           attn = dots.softmax(dim=1)
11 -         attn = (attn + eps) / (attn + eps).sum(dim=-1, keepdim=True)
12           updates = torch.einsum("bjd,bij->bid", v, attn)
13 +         if idx == 0 and self.training:
14 +             var, mean = torch.var_mean(updates, correction=0)
15 +             self.update_buffers(var, mean)
16 +         elif idx == 0 and not self.training:
17 +             var, mean = self.var_buffer, self.mean_buffer
18 +         updates = self.alpha * (updates - mean) / torch.sqrt(var + eps) + self.beta
19           ...
```

---

# H Visual Results

In Tables 6 and 7, we present visual results from object discovery on the CLEVR10 and MOVi-C10 datasets, respectively. On CLEVR, we show both reconstructions and segmentations, while we only provide segmentations on MOVi-C, as the reconstructions are high-dimensional ViT features. In both cases, we perform inference with a high slot count (21).

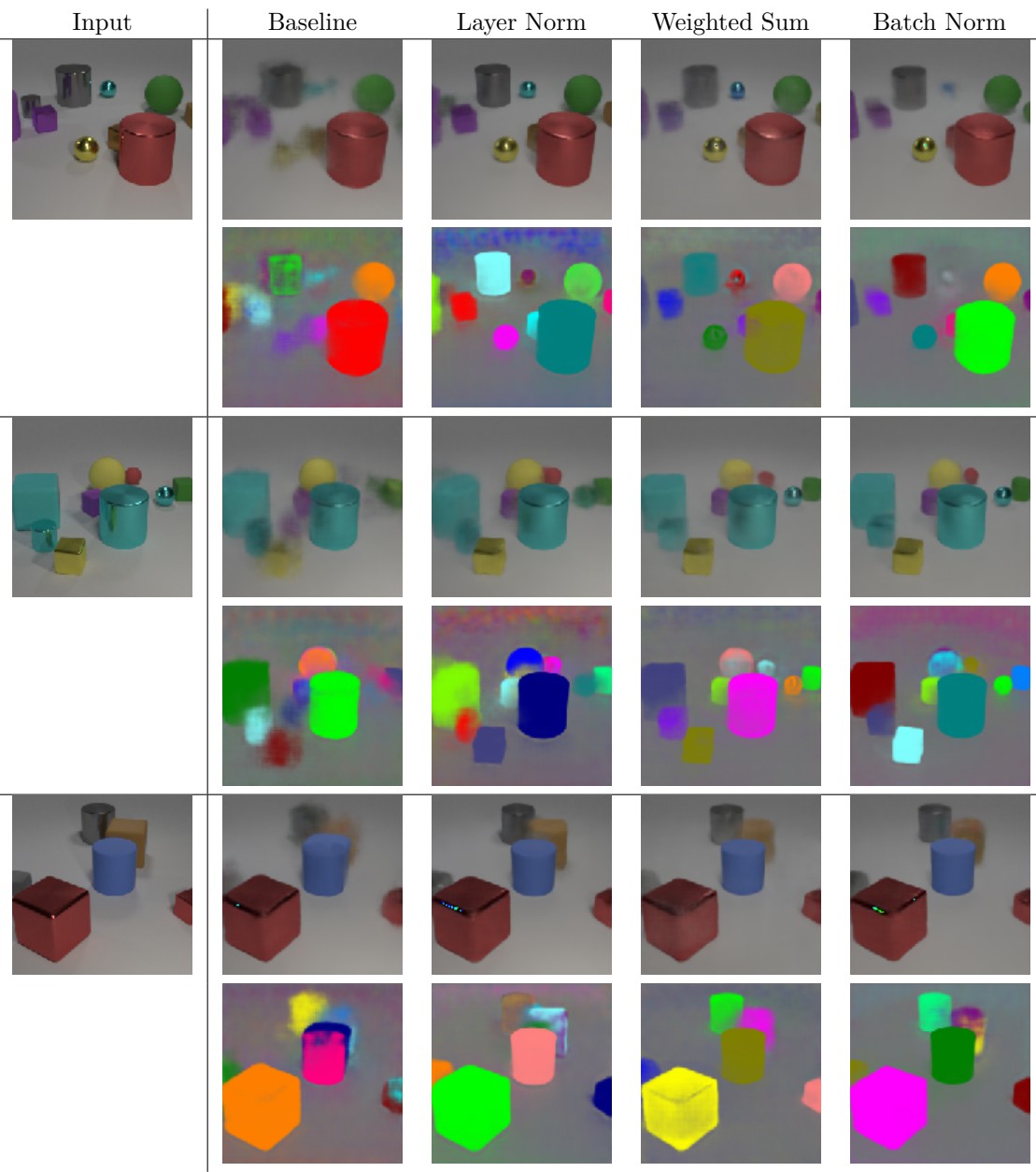

Table 6: Visual results for object discovery on CLEVR10, trained on CLEVR6 with 7 slots. Showing reconstructions and segmentations. Evaluated with 21 slots.

| Baseline | Layer Norm | Weighted Sum | Batch Norm |
|---|---|---|---|

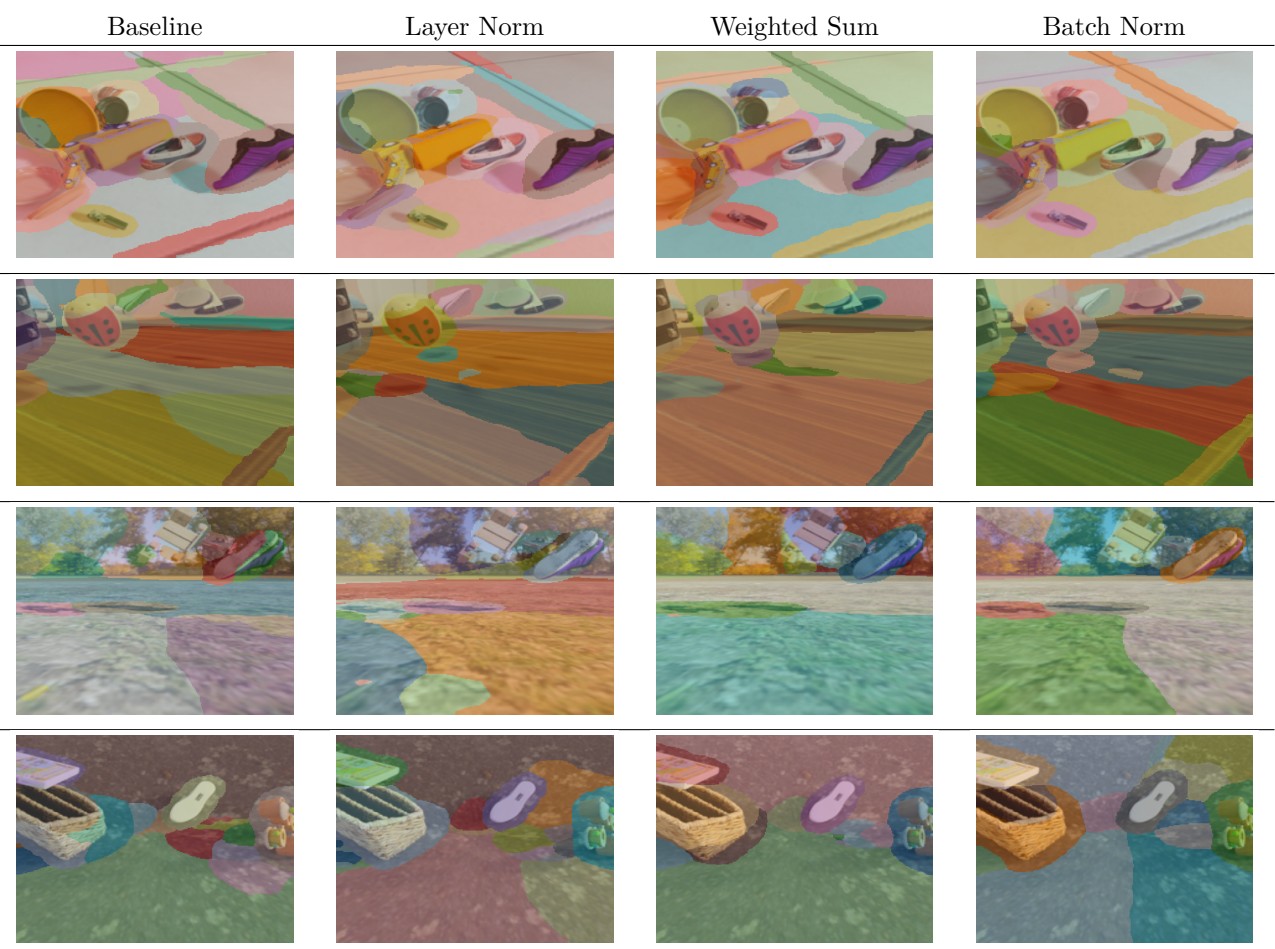

Table 7: Visual results for object discovery on MOViC10, trained on MOViC10 with 11 slots. Evaluated with 21 slots.

# I Property Prediction

We further illustrate the proposed normalizations on a property prediction task on the CLEVR10 dataset. We closely follow the experimental setup detailed by Locatello et al. (2020), using three seeds per variant. We train the models on the CLEVR10 dataset using 10 slots. In Figure 12, we plot how the foreground ARI varies as the number of slots is modified during inference. Consistent with our observations on the object discovery task, we find that the weighted mean and layer norm variants suffer from excess slots, while the proposed normalizations are robust to them. Overall, we find that the batch norm variant performs best w.r.t. foreground segmentation quality.

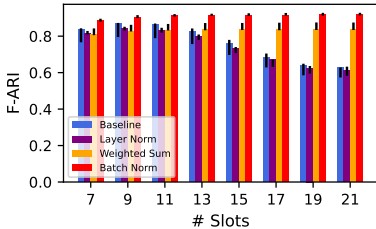

Figure 12: F-ARI (↑) for property prediction on CLEVR10

In Figure 13, we additionally show the mean average prediction at various distance thresholds, as defined in (Locatello et al., 2020). We observe a similar behavior as before, namely that the proposed variants are robust to high slot counts during inference, while the weighted mean and layer norm variants are not. However, we generally find that the weighted mean variant appears to perform best at low slot counts.

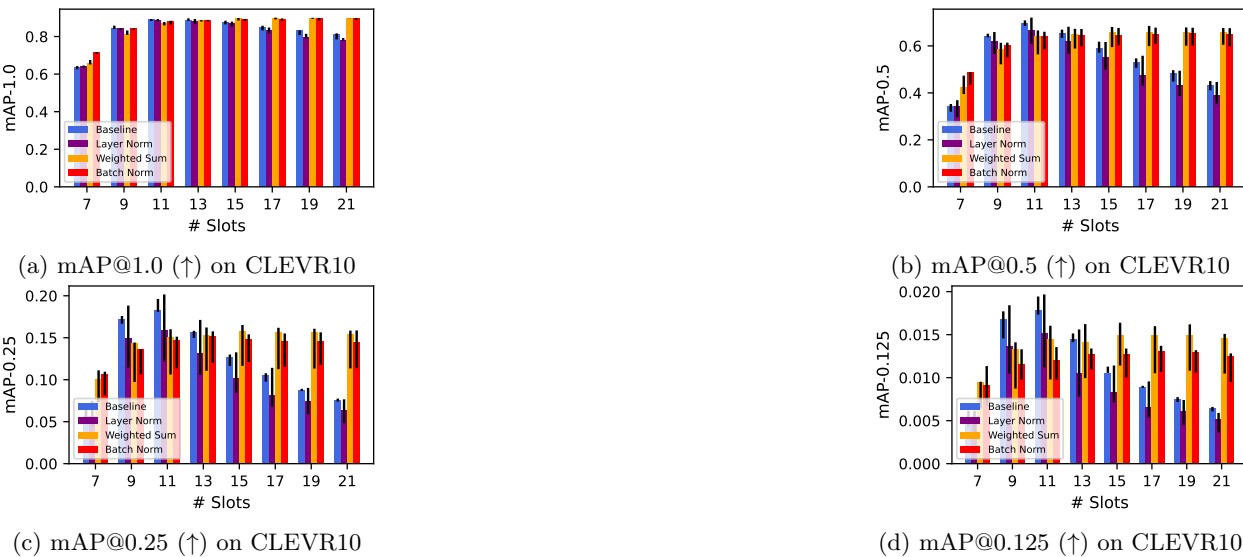

(a) mAP@1.0 (↑) on CLEVR10

(b) mAP@0.5 (↑) on CLEVR10

(c) mAP@0.25 (↑) on CLEVR10

(d) mAP@0.125 (↑) on CLEVR10

Figure 13: Mean average precision at different distance thresholds for property prediction on CLEVR.

## J Ablation of Normalization Schemes

In this section, we study two alternative normalizations methods. Firstly, we consider the weighted sum variant with the scaling parameter chosen as $C = 1$. We refer to this method as "unnormalized". Secondly, we consider a variant which ablates the normalization across the batch axis from the batch normalized variant. I.e., we compute mean and variance for each instance separately across only the slot and layer axes. Hence, we also do not have to keep a moving average of the normalizing statistics for inference. Instead the normalization behaves identically during training and inference. As in the batch normalized variant, two scalar values $\alpha$ and $\beta$ are learned. We refer to this variant as "K-D-Layer Norm", to reflect that we are performing a layer normalization across the slot (K) and layer (D) axes. We perform experiments on the property prediction task on CLEVR with three seeds per variant.

The unnormalized variant fails to obtain object-centric behavior, as becomes apparent from the low foreground ARI across all slot counts, as shown in Figure 14. In Figure 15, we observe that the mean average precision suffers correspondingly. This underscores the importance of scaling the update codes appropriately to achieve competitive performance. While the K-D-Layer Norm performs better, we find that as the other baseline variants, it is not robust to high slot counts. Moreover, on this specific task, it seems to generally underperform in terms of mAP when compared to the other object-centric variants.

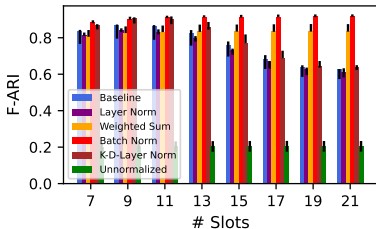

Figure 14: F-ARI (↑) for property prediction on CLEVR10

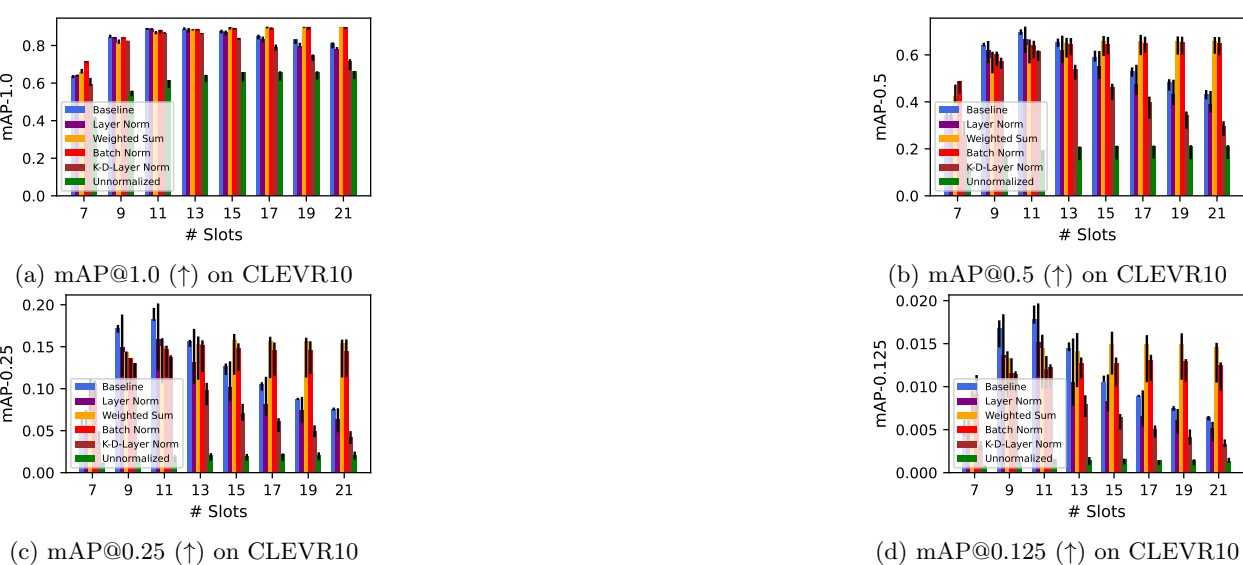

(a) mAP@1.0 (↑) on CLEVR10

(b) mAP@0.5 (↑) on CLEVR10

(c) mAP@0.25 (↑) on CLEVR10

(d) mAP@0.125 (↑) on CLEVR10

Figure 15: Mean average precision at different distance thresholds for property prediction on CLEVR.

