# OpenReview forum: "Attention Normalization Impacts Cardinality Generalization in Slot Attention"
_TMLR — Accepted by TMLR_

### Review · Reviewer_wHLB · 2024-06-24

**Summary Of Contributions:**

- The paper primarily explores the effects of normalisation within the slot-attention architecture. To achieve this, the author suggests employing the Von-Mises-Fisher distribution to model the SA algorithm as an uncoupled iterative EM algorithm.

- Two implementation schemes for the proposed method are provided, each offering distinct advantages.

- The normalisation scheme is finally tested on the CLEVER 6 and MOVi datasets.

**Audience:**

Yes

**Claims And Evidence:**

No

**Requested Changes:**

- The contribution is not clear from the abstract and introduction. The authors discuss different types of normalisation, so introducing the differences in the normalisation schemes earlier would help readers.

- The authors mention the misalignment in EM steps, but its effects have not been discussed in detail. It would be interesting to see the performance differences between aligned and unaligned EM algorithms with vMF distributions, similar to the analysis in [1].

- The concentration parameter $\tau$ in the vMF distribution is somewhat analogous to variance in the normal distribution. How reasonable would it be to make $\tau$ a learnable parameter? This could result in learning the true underlying slot distributions.

- It would be valuable to include an ARD analysis on the mixing coefficients $\pi$.

- Please include visual results, demonstrating improvements in object discovery.


[1] Kori, A., Locatello, F., Santhirasekaram, A., Toni, F., Glocker, B. and Ribeiro, F.D.S., 2024. Identifiable Object-Centric Representation Learning via Probabilistic Slot Attention. _arXiv preprint arXiv:2406.07141_.

**Strengths And Weaknesses:**

- One major issue with SA is determining the number of $K$ during training. The proposed normalisation schemes generalise well even when the number of objects is misspecified during training, which is a very interesting result.

- The proposed method appears to generalise to various numbers of objects, as demonstrated by its zero-shot capabilities.

- Regarding Proposition 1, am I correct in stating that, in the case of SA with mean normalisation, it is impossible to find mixing coefficients $\pi$? If not, what is the implication of Proposition 1?

- For Fixed Scaling normalisation, ablations with $C=[1,<N,N,>N]$ would be useful to convey the point more effectively, given the wide bounds in Equation 13.

- For Batch Scaling, I understand that normalising across the batch index aims to achieve scalar normalisation for a mini-batch. However, how does this approach compare to image-level normalisation (i.e., normalising only along the $K$ and $D$ axes)?

- The authors claim that the SA discards object size information. Could you please provide more details? Given that the decoder reconstructs by generating slots in appropriate positions, it seems that slot representations do include positional knowledge.

- What are the advantages of splitting the subspace $A=a+V$ and under what conditions does this hold true?

---

> ### Author Response · Authors · 2024-07-17
>
> ## Strengths and Weaknesses
> > Regarding Proposition 1...
>
> In Proposition 1 we show that it is not possible to infer the mixing coefficient $\pi_k$ from the update code $u_k$ for all possible settings of slot codes. Specifically, we show that there are some slot codes for which this is not possible. Namely, there are choices of slots such that we obtain the same update code but different mixture coefficients. We clarified this in the paper, changes are marked in red.
>
> > Ablations on $C$
>
> If time permits, we will provide an ablation for $C=1$ on the property prediction task on CLEVR. We are not convinced that we can provide insightful experiments for $C>N$ and $C<N$, as it remains ambiguous how $C$ should be chosen. Even if, say, we were to find that $C=N/2$ provides superior performance on property prediction on CLEVR, we feel that this isolated finding would be insufficient to make any recommendations w.r.t. the choice of $C$. However, we have updated the manuscript to reflect that some tuning of this hyperparameter could be useful.
>
> > Image-level normalization
>
> Thank you for this suggestion! If time permits, we will attempt to follow up with experimental results.
>
> > The authors claim that the SA discards object size information.
>
> We assume you are referring to the statement "We argue that the original Slot Attention normalization scheme discards information on the objects’ sizes" from the abstract. We agree that this is an inaccurate statement and we removed it.
>
> > What are the advantages of splitting the subspace $A=a+V$ and under what conditions does this hold true?
>
> Generally, any affine subspace $A$ can be decomposed in this fashion where $a \in A$  can be arbitrary and $V$ is independent of $a$. We would read this as "$A$ is obtained by translating $V$ along $a$". The decomposition can be made unique by placing additional requirements on $a$, e.g. that it is orthogonal to $V$, which is what we did.
>
> The reason why we introduce this decomposition is that it allows us to easily infer column sums from update codes *in the weighted sum case*. Assume that the values $v_n$ lie in an affine subspace $a + V$ with $a \neq 0$ and $a \perp V$. This is the case iff the affine subspace is not a vector space, i.e. $0 \not\in A$ (we show in the paper that for SA this is almost always true). Then, for the update code $u_k := 1/N\sum_n \gamma_{n, k}v_n$, we can easily infer $1/N\sum_n \gamma_{n, k}$, simply by forming the dot product $\langle a, u_k\rangle$, as this will be exactly $\Vert a\Vert_2^2/N\sum_n \gamma_{n, k}$. As an illustrative example, consider the case where we have $$v_n \in A = \\{ (x_1, ..., x_{d-1}, 1) : x_1, ..., x_{d-1} \in \mathbb{R}\\}$$
> Then, we have $a=(0,\dots,0, 1)$ and it is easy to see that the last coordinate of $u_k$ is exactly $1/N\sum_n \gamma_{n, k}$.
>
>
> ## Requested Changes
> > The contribution is not clear from the abstract and introduction
>
> We have updated the abstract, changes are marked in red.
>
> > The authors mention the misalignment in EM steps, but its effects have not been discussed in detail
>
> We are not certain we understand what you mean by misalignment. Are you referring to the fact that we are using key and value maps and thereby deviate from the scheme that would strictly speaking be dictated by EM? It is our understanding that the authors of [1] refer to this as "decoupling".
>
> Are you proposing that we attempt replacing the SA module with a module that simply performs vanilla vMF clustering without any learnable parameters?
>
> > The concentration parameter $\tau$ the vMF distribution is somewhat analogous to variance in the normal distribution. How reasonable would it be to make $\tau$ a learnable parameter?
>
> In our exposition of EM in vMF mixture models, we have assumed $\tau$ to be fixed and identical for all components. In this setting, the temperature $\frac{1}{\sqrt D}$ in the softmax of SA acts analogously to the concentration parameter in vMF distributions. While this temperature parameter could be turned into a learnable parameter, that would not add any capacity to the model, as keys and queries are already produced via learnable linear maps.
>
> > It would be valuable to include an ARD analysis on the mixing coefficients
>
> Are you referring to an approach similar to the one in [1] (i.e. pruning slots with low mixing coefficients)? Furthermore, are you referring to a post-hoc analysis on the trained models or would you be interested in training with ARD?
>
> > Please include visual results, demonstrating improvements in object discovery
>
> We have added visualizations for object discovery on CLEVR and MOVi-C with excess slots during inferences to the appendix.
>
>
>
> [1] Kori, A., Locatello, F., Santhirasekaram, A., Toni, F., Glocker, B. and Ribeiro, F.D.S., 2024. Identifiable Object-Centric Representation Learning via Probabilistic Slot Attention.
>
> [2] Kirilenko, D., Kovalev, A., & Panov, A. (2023). Object-Centric Learning with Slot Mixture Models.

---

> > ### Author Response · Authors · 2024-07-28
> >
> > We have added some of the requested experimental results on the scaling parameter $C$ and the "image-level normalization" to the appendix.
> > We evaluate the property prediction task on CLEVR and find that with $C=1$, the models fail to obtain object-centric behavior.
> > Regarding image-level normalization, we find that while the variant yields object-centric behavior, it is not robust to high slot counts.

---

> > > ### Comment · Reviewer_wHLB · 2024-08-05
> > >
> > > Thank you for the thoughtful rebuttal.
> > >
> > > The rebuttal has addressed most of my concerns. Regarding EM decoupling, ideally, both the E and M steps should have the same parameters. However, due to the projections, the parameters are decoupled. I am interested in seeing the effect of removing projections and simply performing vMF clustering.
> > >
> > > Additionally, for ARD, it would be helpful to visualize the contribution of the mixing coefficients. Printing the weightage of the mixing coefficients over each slot would be particularly useful given that the visual results are a bit noisy.

---

### Review · Reviewer_433y · 2024-06-25

**Summary Of Contributions:**

Slot attention in object-centric learning is a way to bind different "slots" to different objects in the scene. It involves an attention mechanism over different slots. This work proposes simple changes to slot attention, in particular, to the normalization scene of the attention vectors. The authors show that the current slot-attention normalization of the weighted mean of slot vectors is one factor that significantly impedes the generalization of slot-attention to OOD number of objects. To remedy this, authors propose 2 extensions of the normalization -- weighted sum and batch norm for attention vectors and show how these two normalization schemes can generalize to OOD number of slots, number of objects, and even OOD appearance in CLEVR and MOVi-C datasets.

**Audience:**

Yes

**Broader Impact Concerns:**

None required.

**Claims And Evidence:**

Yes

**Requested Changes:**

1. Why is it an appropriate choice to model the slots via a mixture of von Mises-Fisher distributions? I believe the manuscript needs a paragraph on what is it about this distribution that makes it amenable to object-centric slot attention-based methods.

2. Section 3.1 also needs to cite and briefly discuss [1]. [1] discusses the relation of EM on mixture of von Mises-Fisher distributions.

3. A pseudocode of the proposed normalization scheme similar to the slot-attention paper. In particular, the changes (highlighted in a different color) from the original slot-attention algorithm would be *incredibly* beneficial to readers who are already familiar with slot-attention.

4. Please add visual results of the CLEVR and MOVI-C scenes, especially in the cases where there is a significant difference during inference with a higher number of slots.

5. I'd also request the authors to address my **weaknesses** points appropriately with experiments for #1 and text for #2.

**Strengths And Weaknesses:**

**Strengths**

1. A simple normalization scheme on top of slot-attention to make it more robust to increase the number of slots during inference.

2. Extensive and insightful experiments on object-centric video datasets.

**Weaknesses**

1. My major concern with this work is the lack of experimentation on complex scenes. In particular, even within the MOVi datasets, there are MOVi-D and E datasets that are more complex than MOVi-C in terms of camera motion and textured background so I'm curious as to how the results shown in 5 (a-b) generalize to more complex scenes. In addition, it would be good to see if there is any benefit of the proposed weighted-sum normalization over attention in the case of CATERTex [2] and MOVi-Tex scenes as well.


2. The related works section needs to be elaborated and is poorly written in terms of lack of enough citations of prior object-centric works [7, 8] and prior slot-attention based works [3, 4, 5, 6] (I've only cited a few -- please refer to these works and their related works section to see more references). I find the related works section related to the normalization aspect of slot vectors to be sufficient however, the purpose of the related works section is to situate your work among the pool of existing works, and contrasting the aforementioned works would help the reader to gain some context of this work.

----

**References**

[1] Generative Model-based Clustering of Directional Data, Arindam Banerjee et al., KDD 2003

[2] CATER: A diagnostic dataset for Compositional Actions and TEmporal Reasoning, Giridhar et al., ICLR 2020

[3] Slotformer: Unsupervised visual dynamics simulation with object-centric models, Wu et al., ICLR 2023

[4] Conditional Object-Centric Learning from Video, Kipf et al., ICLR 2022

[5] Invariant Slot Attention: Object Discovery with Slot-Centric Reference Frames, Biza et al., ICML 2023

[6] Simple unsupervised object-centric learning for complex and naturalistic videos, Singh et al., NeurIPS 2022

[7] Space: Unsupervised object-oriented scene representation via spatial attention and decomposition, Lin et al., ICLR 2020

[8] Spatially invariant unsupervised object detection with convolutional neural networks, Crawford & Pineau, AAAI 2019

---

> ### Author Response · Authors · 2024-07-17
>
> Thank you for your thoughtful and constructive feedback! We address your requested changes and weaknesses below.
> > Why is it an appropriate choice to model the slots via a mixture of von Mises-Fisher distributions?
>
> Many works (as cited in sec 3) already draw a comparison between SA and EM in Gaussian mixture models. This comparison is lacking in that EM in GMMs would require a comparison of cluster centers (slots) to data points (image features) via euclidean distance, while SA compares the two via dot product similarity.
> Alleviating this discrepancy is our main motivation for transitioning from a GMM to a mixture of vMF distributions. Additionally, we note that the layer normalization of image features and slots resembles the $\ell^2$ normalizations performed in vMF mixtures.
> These motivations are also noted briefly at the beginning of section 3.
>
> > Citation for EM on vMF mixtures
>
> Thank you for this pointer, we have updated the manuscript to cite this work. However, since vMF clustering is simply a specific instance of the well-known EM algorithm, we believe that no detailed discussion of [1] is required.
>
> > pseudocode of the proposed normalization scheme similar to the slot-attention paper.
>
> We have added pseudocode in a diff format to the appendix.
>
> > Please add visual results of the CLEVR and MOVI-C scenes
>
> We have added visualizations for inference with 21 slots to the appendix.
>
> Regarding the weaknesses you have pointed out:
>
> 1. It is our understanding from https://github.com/google-research/kubric/tree/main/challenges/movi that MOVi-E and MOVi-F deviate from MOVi-D only in that camera motion and motion blur are added. Since we perform object discovery on still images and do not use any motion cues, we argue that MOVi-D is just as complex as MOVi-E and MOVi-F in our setting. While we do not train on MOVi-D, we do perform zero-shot evaluations on that dataset. MOVi-C, on which we do train, differs from MOVi-D only in that fewer objects are present in the scene. However, the objects themselves are identical (highly textured and 3D-scanned) and the background is a real-world HDRI. Generally, we would strongly argue that, at this time, MOVi-C is among the most realistic and complex datasets for object discovery on still images.
>
>     Following you suggestion, we have taken the CaterTex and MOViTex datasets under consideration. However, we found that [6] states:
>
>     > We conjecture that due to the similar texture in the background and foreground, inferring correct object regions from a static image can be harder than inferring them when the objects are moving. This may explain the significantly larger gap with SLATE in MOVi-Tex compared to the other datasets. Note that this gap exists even when evaluating STEVE with zero past frames (i.e. a static image), indicating that training on temporal data helps STEVE infer segments on static images.
>
>     While the mentioned performance gap is less substantial in CATERTex, it is still noticable. We study the setting of object discovery in still images and do not take motion cues into account. Hence, we do not believe that the MOVi-Tex and CATER-Tex datasets would be appropriate benchmarks in our setting.
>
> 2. Thank you for these suggestions. We have revised the related work section accordingly.

---

> > ### Comment · Reviewer_433y · 2024-08-01
> > **Thanks for the clarification**
> >
> > Thanks to the authors for their rebuttal and clarifications. I concur to the authors that their work is primarily in the space of static camera observations and CATER-Tex and MoVi-Tex aren't the best datasets to perform experiments on.
> >
> > Thanks for elaborating the related works, and adding the pseudo-code (it's now quite easier to read and find out what are the core changes in terms of normalization as compared to the original Slot Attention).
> >
> > Also, the results of property prediction as suggested by _Reviewer 48Hx_ further corroborate the hypothesis of this paper.
> >
> > I've provided my official recommendation to the Action Editor.

---

### Review · Reviewer_48Hx · 2024-07-04

**Summary Of Contributions:**

This paper investigates the normalization scheme in Slot Attention. The authors draw connections between Slot Attention and expectation maximization in a mixture model of von Mises-Fisher distributions and argue that weighted sum normalization preserves information on the fraction of input tokens assigned to the slots. In experiments, they demonstrate that weighted sum normalization and batch normalization show better generalization capability when scaling to varying number of slots during inference when compared to weighted mean normalization or layer normalization.

**Audience:**

Yes

**Claims And Evidence:**

Yes

**Requested Changes:**

I’ve listed several suggestions in the Weaknesses / Questions section above. While I believe addressing all of them would make a stronger paper, addressing #2 is critical since it is not supported in the paper. I am also especially curious about #5.

**Strengths And Weaknesses:**

**Strengths**

1. This paper is well-motivated in studying the generalization capabilities of Slot Attention. In particular, the normalization aspect of Slot Attention has not been scrutinized in prior research.
2. The connection between the Slot Attention algorithm and EM in a mixture model of von Mises-Fisher distributions is interesting and seems sound, from my understanding.
3. The experiments thoroughly study how varying the number of slots and objects affects segmentation performance across several datasets and methodologies, although I believe there could be more investigations into other aspects of the slot representations, as I describe below.

**Weaknesses / Questions**

1. Currently, all evaluations in the paper are with regard to segmentation quality. It would be very informative to investigate the quality of the representations by property prediction (e.g. as is done in [1, 2]) under the different normalization schemes. I understand that many recent works on object-centric learning only report FG-ARI, but in proposing a new normalization scheme, it is important to see its effect on the learned representation.
2. In the Abstract, the authors write: “We argue that the original Slot Attention normalization scheme discards information on the objects’ sizes, which impairs its generalization capabilities.” However, object size is never investigated in the paper.
3. The authors mention [3] as a related work that also investigates normalization in Slot Attention, but do not compare with it in their experiments. I believe at least comparing with SA-SH from their paper would be informative.
4. The error bars seem larger with weighted sum normalization and the authors also state that for two seeds, the autoencoders decompose the input spatially instead of object-wise. Did the authors generally find these normalization schemes to be less stable to train?
5. From Proposition 1, it seems one of the main takeaways is that with the weighted mean normalization that is normally done in Slot Attention, the column sums in the attention matrix will be different if the number of slots is changed at inference time. This is supported by their experiments showing worse segmentation quality when the number of slots during inference is different than the number of slots during training. It also makes sense intuitively because the scaling is different at inference time than at training time. What would happen if we use weighted mean normalization but scale the attention matrix by num_slots_inference / num_slots_training when evaluating with num_slots_inference?

[1] Object-Centric Learning with Slot Attention (https://arxiv.org/abs/2006.15055)

[2] Object-Centric Slot Diffusion (https://arxiv.org/abs/2303.10834)

[3] Unlocking Slot Attention by Changing Optimal Transport Costs (https://arxiv.org/abs/2301.13197)

---

> ### Author Response · Authors · 2024-07-17
>
> Thank you for your thoughtful feedback! We discuss the points you raised in the following.
> 1. In the context of object discovery, we consider FG-ARI to be the main metric, as we interpret it as a measure of object-centricness of the learned model. Notwithstanding, we concur that other task-specific performance measures are also relevant in assessing the proposed method. We include plots on reconstruction losses and overall ARI (including background pixels) for object-discovery in the appendix. Following your request, we also implemented experiments for property prediction and provide the results (as measure via F-ARI and mAP at different distance thresholds) in the appendix.
> 2. We agree that the statement in the abstract is inaccurate and we removed it.
> 3. Unfotunately, we expect that we will not have sufficient time or computational resources to perform such experiments. While an empirical investigation of SA-SH could complement our current work, we do not feel that it would be essential to it. Our goal is to demonstrate that *minimal* modifications (which are easily adoptable by other researchers and practitioners) can lead to improved generalization. SA-SH, in contrast to our methods, introduces substantial algorithmic complexity.
> 4. Spatial decomposition failures were only observed on the CLEVR dataset with the weighted sum or batch norm variant. Overall, it is difficult to accurately attribute the causes of these failures. Anecdotally, we report that the complexity of the dataset (and the corresponding architecture) has a large impact on the prevalence of failures with the simple tetrominoes dataset being most prone to it (independently of the normalization method) and no failures occuring at all on the MOVi datasets. Overall, our normalizations seemed just as stable as the baselines on the MOVi dataset but perhaps more prone to spatial decompositions on CLEVR. In cases where this is an issue, we hypothesize that  small modifications (e.g. learning rate warmup as noted in the original SA paper or other approaches like implicit SA [1]) can be sufficient to rescue stability. We note that we did not tune hyperparameters to optimize stability for our methods and instead used the settings proposed in previous works.
> 5. Thank you for this suggestion! While we believe we understand your idea, we would like to ask you to clarify this a bit before starting experiments. In our notation, simply scaling the attention matrix $\Gamma$ by a scalar before performing mean normalization will effectively not change the computational graph, as the scalar will appear both in the numerator and the denominator of equation 4. Are you perhaps instead suggesting to modify equation 4 to something like:
> $$ u_k := \frac{\text{num train slots}}{\text{num inference slots}}\frac{\tilde u_k}{\sum_n\gamma_{n, k}}$$
> If time permits, we will attempt to follow up with an evaluation.
>
> [1] Chang, M., Griffiths, T., & Levine, S. (2022). Object Representations as Fixed Points: Training Iterative Refinement Algorithms with Implicit Differentiation. In S. Koyejo, S. Mohamed, A. Agarwal, D. Belgrave, K. Cho, & A. Oh (Eds.), Advances in Neural Information Processing Systems 35: Annual Conference on Neural Information Processing Systems 2022, NeurIPS 2022, New Orleans, LA, USA, November 28 - December 9, 2022. Retrieved from http://papers.nips.cc/paper_files/paper/2022/hash/d301e2878a7ebadf1a95029e904fc7d0-Abstract-Conference.html

---

> > ### Comment · Reviewer_48Hx · 2024-07-28
> > **Response**
> >
> > Thank you for updating the paper. The property prediction results look convincing.
> >
> > 5. I think I may have misunderstood Proposition 1 on the first reading. After looking at it again, I'm not sure if my original suggestion makes much sense anymore, since as you mention both numerator and denominator would be scaled by the same factor.

---

### Decision · Action_Editor_Ruet · 2024-08-19

**Recommendation:** Accept as is

**Comment:**

The reviewers were convinced by the paper and the authors' responses to their initial comments/questions.

**Audience:**

Yes

**Claims And Evidence:**

Yes